# Meta-analysis shows the impacts of ecological restoration on greenhouse gas emissions

Tiehu He [1,2,3,4], Weixin Ding [5], Xiaoli Cheng [6], Yanjiang Cai[7], Yulong Zhang[8], Huijuan Xia[1,2], Xia Wang[1,2], Jiehao Zhang[1,2], Kerong Zhang [1,2,3,4] ✉ & Quanfa Zhang[1,2]

International initiatives set ambitious targets for ecological restoration, which is considered a promising greenhouse gas mitigation strategy. Here, we conduct a meta-analysis to quantify the impacts of ecological restoration on greenhouse gas emissions using a dataset compiled from 253 articles. Our findings reveal that forest and grassland restoration increase $CH_4$ uptake by 90.0% and 30.8%, respectively, mainly due to changes in soil properties. Conversely, wetland restoration increases $CH_4$ emissions by 544.4%, primarily attributable to elevated water table depth. Forest and grassland restoration have no significant effect on $N_2O$ emissions, while wetland restoration reduces $N_2O$ emissions by 68.6%. Wetland restoration enhances net $CO_2$ uptake, and the transition from net $CO_2$ sources to net sinks takes approximately 4 years following restoration. The net ecosystem $CO_2$ exchange of the restored forests decreases with restoration age, and the transition from net $CO_2$ sources to net sinks takes about 3-5 years for afforestation and reforestation sites, and 6-13 years for clear-cutting and post-fire sites. Overall, forest, grassland and wetland restoration decrease the global warming potentials by 327.7%, 157.7% and 62.0% compared with their paired control ecosystems, respectively. Our findings suggest that afforestation, reforestation, rewetting drained wetlands, and restoring degraded grasslands through grazing exclusion, reducing grazing intensity, or converting croplands to grasslands can effectively mitigate greenhouse gas emissions.

Global temperature is approaching a threshold that will have irreversible consequences for the future of our Earth, mainly due to the increasing concentrations of atmospheric greenhouse gases (GHG) such as carbon dioxide ($CO_2$), methane ($CH_4$) and nitrous oxide ($N_2O$)[1].

Over the last 220 years, global $CO_2$ concentrations increased from 283 to 419 parts per million (ppm), $CH_4$ increased from 750 to 1925 parts per billion (ppb), and $N_2O$ increased from 273 to 336 ppb[2]. Land-use change and ecosystem degradation have caused massive

[1]Key Laboratory of Aquatic Botany and Watershed Ecology, Wuhan Botanical Garden, Chinese Academy of Sciences, Wuhan 430074, P.R. China. [2]Danjiangkou Wetland Ecosystem Field Scientific Observation and Research Station, the Chinese Academy of Sciences & Hubei Province, Wuhan 430074, P.R. China. [3]Key Laboratory of Lake and Watershed Science for Water Security, Nanjing Institute of Geography and Limnology, Chinese Academy of Sciences, Nanjing 210008, China. [4]Hubei Key Laboratory of Wetland Evolution & Ecological Restoration, Wuhan Botanical Garden, Chinese Academy of Sciences, Wuhan 430074, China. [5]State Key Laboratory of Soil and Sustainable Agriculture, Institute of Soil Science, Chinese Academy of Sciences, Nanjing 210008, China. [6]School of Ecology and Environmental Science, Yunnan University, Kunming 650091, P. R. China. [7]State Key Laboratory of Subtropical Silviculture, Zhejiang A&F University, Hangzhou 311300, China. [8]Eastern Forest Environmental Threat Assessment Center, Southern Research Station, USDA Forest Service, Research Triangle Park, NC 27709, USA. ✉e-mail: kerongzhang@wbgcas.cn

anthropogenic emissions of GHG and altered natural ecological ecosystems from net sinks to net sources[3,4]. Restoring the degraded ecosystems and converting lands back to healthy ecosystems has been proposed as a vital strategy for stabilizing the Earth's climate[5]. To limit global warming below the 2 °C threshold, there is an urgent need to reduce atmospheric GHG concentrations by restoring degraded ecosystems such as forests, grasslands and wetlands[6]. Ecological restoration is the process of assisting the recovery of an ecosystem that has been degraded, damaged, or destroyed (Society for Ecological Restoration and Policy Working Group 2002). The United Nations (UN) has declared 2021-2030 as the 'UN Decade on Ecosystem Restoration' and calls on countries to meet commitments to restore one billion hectares of land. The Bonn Challenge and the New York Declaration on Forests have established ambitious targets to restore 350 million hectares of forests worldwide by 2030[5]. Thus, systematically understanding the impacts of ecological restoration on GHG emissions is imperative for making better restoration policies and improving the Intergovernmental Panel on Climate Change (IPCC) guidance of GHG inventories.

Forests occupy approximately 30% of the global land surface and play a crucial role in regulating the global carbon (C) cycle and reducing global warming[7–9]. A recent estimation reported that global forests maintained a net C sink of −7.6 Gt $CO_2$e $yr^{-1}$, reflecting a balance between gross C removals (−15.6 Gt $CO_2$e $yr^{-1}$) and gross emissions (8.1 Gt $CO_2$e $yr^{-1}$) from deforestation and other disturbances, e.g., clear-cut, fire, windthrows, insects, etc[10]. Afforestation and reforestation could change biomass accumulation and alter soil biogeochemical, physical and hydrological properties, thereby affecting the GHG fluxes[9,11–13]. Previous work found that converting croplands to forests increased $CH_4$ uptake due to the decreased soil bulk density, and afforestation decreased $N_2O$ emissions due to the reduced nitrogen (N) substrate availability[11]. The conversion of grasslands to forests might decrease $CH_4$ emissions but increase $N_2O$ emissions[12]. Although many studies showed that afforestation could enhance the $CO_2$ sink function of ecosystems[10,14], some studies observed that forest lands continued to act as a $CO_2$ source even after several years of afforestation[15]. These diverse results suggest that the magnitude and direction of GHG dynamics driven by forest restoration are highly uncertain and could be affected by multiple factors, including ecosystem types, restoration ways, and restoration age[12,16,17]. It is undoubtedly necessary to explore the general patterns and the major controlling factors of GHG emissions in the restored forests.

Grassland ecosystems constitute approximately 40% of the terrestrial biosphere[18], and natural grasslands are usually identified as efficient sinks of atmospheric $CH_4$ and $CO_2$[19], but sources of $N_2O$[20]. Grassland degradation leads to changes in soil nutrient content, soil moisture, and plant composition, which influences the pattern of GHG emissions[21,22]. It has been found that grassland degradation might decrease $CH_4$ uptake by 40%[23]. However, whether grassland restoration can reduce GHG emissions is still inconclusive[19,24]. Previous work reported that grassland restoration increased C accumulation and enhanced $CH_4$ uptake[24], but some studies found that grassland restoration might stimulate $N_2O$ and $CO_2$ emissions and shift grassland from a C sink to a C source[19]. Furthermore, the effects of grassland types, restoration measures, and restoration age on GHG emissions in the restored grasslands at a global scale are still unclear.

Wetlands are considered to be one of the most efficient ecosystems for sequestrating $CO_2$ from the atmosphere[25], mainly because inundation creates anaerobic conditions that prevent the decomposition of dead plant material and restore sequestered C in soil[26,27]. Despite covering only 5–8% of the Earth's landscape, global wetlands store 20–30% of soil C on the Earth and thereby play an important role in the global C cycle[28]. In general, wetland drainage and degradation decrease $CH_4$ emissions but enhance $CO_2$ and $N_2O$ emissions to the atmosphere, converting the wetlands from C sinks into sources[29,30]. However, the impacts of restoration on wetland GHG and the driving factors remain controversial[31–33]. Previous work reported that wetland restoration could shift the ecosystems into net GHG sources[32,34,35] or net sinks[36]. The inconsistent results are probably attributed to the wetland restoration types, restoration age, climate, water table depth, and soil properties[32,37]. Since wetland restoration generally decreases $CO_2$ emissions but increases $CH_4$ emissions[32,34,36], the overall effects of wetland restoration on the global warming potentials (GWP) considering three major GHGs (i.e., $CO_2$, $CH_4$, and $N_2O$) are not well understood.

Despite numerous studies investigating the effects of ecological restoration on the emission of individual or a few GHGs at the plot or regional level[38], the general pattern of the impacts of ecological restoration on the three major GHGs at a global scale has not yet been analyzed. Furthermore, there is currently a lack of comprehensive global assessments for the three major ecosystems (i.e., forests, grasslands, and wetlands) which are crucial for the global GHG budget and the 'UN Decade on Ecosystem Restoration'[6–9]. In addition, detailed data on the responses of GHG to ecological restoration are lacking in the IPCC reports, the IPCC Guidelines for National Greenhouse Gas Inventories, and the Good Practice Guidance for Land Use, Land-Use Change and Forestry. To fill these knowledge gaps, we compiled a global dataset from 253 peer-reviewed articles and conducted a meta-analysis to assess the effects of ecological restoration on GHG emissions (Fig. 1). Our specific objectives were to (1) quantify the impacts of ecological restoration on $CH_4$ and $N_2O$ emissions and net ecosystem $CO_2$ exchange (NEE) in forest, grassland and wetland ecosystems, (2) explore the patterns of GHG emissions with restoration age, and (3) determine the key factors influencing the response of GHG emissions to ecological restoration.

In this work, we show that forest and grassland restoration increases $CH_4$ uptake, mainly due to the changes in soil properties. Conversely, wetland restoration increases $CH_4$ emissions, primarily attributed to elevated water table depth. Forest and grassland restoration has no significant effect on $N_2O$ emissions, while wetland restoration reduces $N_2O$ emissions. Overall, forest, grassland and wetland restoration enhances C sink, reduces the global warming potentials, and can serve as strategies for mitigating GHG.

## Results
### Effects of ecological restoration on $CH_4$ emissions
Overall, forest and grassland restoration significantly decreased $CH_4$ emissions, and the weighted response ratios (RRd) of $CH_4$ emissions were –2.3 (95% CI: –2.9 to –1.6) and –1.6 (95% CI: –2.4 to –0.8) under forest and grassland restoration, respectively (Fig. 2a). Compared with the paired control ecosystems, forest and grassland restoration averagely increased $CH_4$ uptake from 1.0 to 1.9 kg C $ha^{-1}$ $year^{-1}$ (by 90.0%) and 2.6 to 3.4 kg C $ha^{-1}$ $year^{-1}$ (by 30.8%), respectively (Fig. 3b, c). Among the types of grassland restoration, temperate steppe & meadow and desert steppe increased $CH_4$ uptake from 2.6 to 3.8 kg C $ha^{-1}$ $year^{-1}$ (by 46.2%) and 7.7 to 11.4 kg C $ha^{-1}$ $year^{-1}$ (by 48.4%), respectively (Fig. 3c). Wetland restoration significantly increased $CH_4$ emissions by 544.4% (RRd: 2.9; 95% CI: 2.4–3.4; $P < 0.05$) (Fig. 2a). The average $CH_4$ emissions increased from 23.4 kg C $ha^{-1}$ $year^{-1}$ to 150.8 kg C $ha^{-1}$ $year^{-1}$ after wetland restoration (Fig. 3a). Among the types of wetland restoration, the conversion of grasslands to wetlands showed the largest increase in $CH_4$ emissions, with an average increase from 61.2 kg C $ha^{-1}$ $year^{-1}$ (in paired control) to 284.8 kg C $ha^{-1}$ (in restored wetlands) (Fig. 3a). In contrast, there was no significant change in $CH_4$ emissions when aquaculture ponds were converted to wetlands (RRd: –1.7; 95% CI: –3.8 to 0.5) and mangroves were restored (RRd: 1.1; 95% CI: –0.4 to 2.5) (Fig. 2a).

### Effects of ecological restoration on $N_2O$ emissions
Overall, forest (RRd: –0.4; 95% CI: –1.3 to 0.4) restoration did not affect $N_2O$ emissions, while grassland and wetland restoration reduced $N_2O$ emissions by 21.7% (RRd: –0.7; 95% CI: –1.4 to –0.1) and 68.6% (RRd: –2.9;

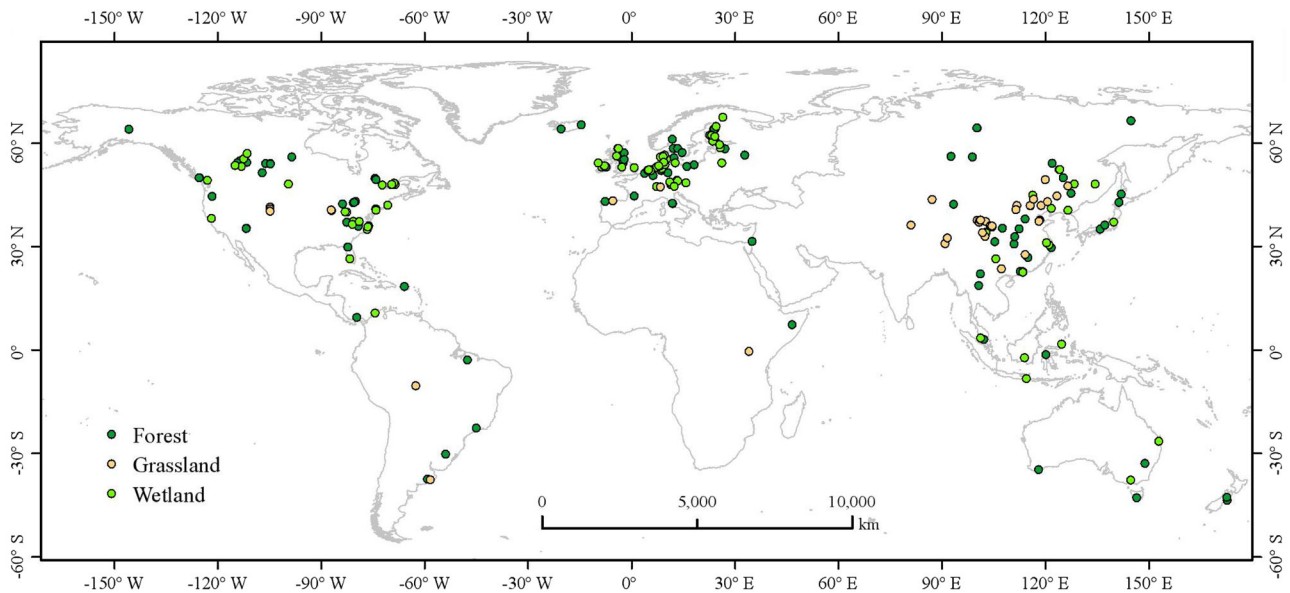

**Fig. 1 | Global distribution of the study sites for this meta-analysis.** The free continental data of the world map was sourced from Natural Earth, supported by the North American Cartographic Information Society (https://www. naturalearthdata.com/). ArcGIS Desktop 10.8 (Esri, West Redlands, CA, USA) was employed for mapping the distribution of the study sites. Source data are provided as a Source Data file.

95% CI: −3.9 to −1.9; $P < 0.05$) (Fig. 2b), respectively. When considering the types of forest restoration, the conversion of croplands to forests averagely decreased $N_2O$ emissions from 3.7 kg N ha$^{-1}$ year$^{-1}$(in paired control) to 1.4 kg N ha$^{-1}$ year$^{-1}$ (in restored forests) (*RRd*: −3.3; 95% CI: −4.7 to −1.9; $P < 0.05$) (Fig. 3e). Among the types of wetland restoration (Fig.2b), the conversion of grasslands to wetlands averagely reduced $N_2O$ emissions from 5.2 kg N ha$^{-1}$ year$^{-1}$ (in paired control) to 2.6 kg N ha$^{-1}$ year$^{-1}$ (in restored wetlands), and the conversion of croplands to wetlands averagely decreased $N_2O$ emissions from 17.0 kg N ha$^{-1}$ year$^{-1}$ to 2.3 kg N ha$^{-1}$ year$^{-1}$. Compared with the paired control ecosystems, peatland restoration averagely reduced $N_2O$ emissions from 2.2 kg N ha$^{-1}$ year$^{-1}$ to 0.5 kg N ha$^{-1}$ year$^{-1}$. However, floodplains restoration did not significantly affect $N_2O$ emissions (Fig. 2b). Among the measures of grassland restoration, the conversion of croplands to grasslands averagely decreased $N_2O$ emissions from 2.3 kg N ha$^{-1}$ year$^{-1}$ to 0.7 kg N ha$^{-1}$ year$^{-1}$ (Supplementary Fig. S1b). Similarly, prairie restoration reduced $N_2O$ emissions from 4.8 kg N ha$^{-1}$ year$^{-1}$ to 0.1 kg N ha$^{-1}$ year$^{-1}$ (*RRd*: −10.9; 95% CI: −14.7 to −7.1; $P < 0.05$) (Fig. 3f).

**Effects of ecological restoration on $CO_2$ fluxes and GWP**

Overall, wetland restoration significantly reduced NEE by 138.8% (*RRd*: −3.2; 95% CI: −3.8 to −2.5; $P < 0.05$) (Figs. 2c and 4a, Table 1). Compared with the paired control ecosystems, the conversion of grasslands to wetlands averagely reduced NEE from 231.9 g C m$^{-2}$ year$^{-1}$ to −219.5 g C m$^{-2}$ year$^{-1}$, and the conversion of aquaculture to wetlands averagely reduced NEE from −41.9 g C m$^{-2}$ year$^{-1}$ to −151.5 g C m$^{-2}$ year$^{-1}$(Supplementary Table S1). Bogs restoration averagely reduced NEE from 159.2 to −35.8 g C m$^{-2}$ year$^{-1}$ (Fig. 4a). The conversion of grasslands to wetlands decreased gross primary productivity (GPP) and ecosystem respiration (ER), while bogs restoration increased GPP and ER (Supplementary Fig. S2). The floodplains and mangrove restoration showed no significant effect on GPP and ER (Supplementary Fig. S2).

Overall, grassland restoration decreased NEE by 146.9% (*RRd*: −4.7; 95% CI: −5.8 to −3.5; $P < 0.05$) (Figs. 2c, 4h and Supplementary Fig. S1c). Compared with the paired control ecosystems, grassland restoration by grazing exclusion averagely decreased NEE from −245.3 g C m$^{-2}$ year$^{-1}$ to −703.0 g C m$^{-2}$ year$^{-1}$, grassland restoration by reducing grazing density averagely reduced NEE from −587.9 g C m$^{-2}$ year$^{-1}$ to −1460.1 g C m$^{-2}$ year$^{-1}$, and the conversion of cropland to grassland

averagely reduced NEE from 10.3 g C m$^{-2}$ year$^{-1}$ to −75.8 g C m$^{-2}$ year$^{-1}$ (Supplementary Fig. S1c, Table S2). Grassland restoration increased GPP and ER (Supplementary Fig. S2).

Due to the small sample size for the paired restored-control measurements for the NEE, GPP, and ER in forests, the effects of forest restoration on $CO_2$ fluxes were not tested by the *RRd* and t-test (Figs. 2c, 4d–f). Based on the restoration chronosequence sub-dataset, the NEE in restored forests decreased first and then tended to be stable and showed a negative exponential relationship with afforestation age, while the GPP and ER showed a positive exponential relationship with afforestation age (Fig. 5e; $P < 0.001$). Similarly, the NEE was negatively and exponentially correlated with reforestation age and time since restoration after disturbance (Fig. 5f, g; $P < 0.001$).

On average, the C budget ($CO_2$ and $CH_4$) was −295.5, −506.5 and −53.4 g C m$^{-2}$ year$^{-1}$ for forest, grassland and wetland restoration, respectively, indicating the capacity of enhanced C sink in the restored ecosystems (Table 1). On average, forest, grassland and wetland restoration decreased the GWP by 327.7%, 157.7% and 62.0% compared with their paired control ecosystems, respectively (Table 1).

**Changes of $CH_4$ and $N_2O$ emissions and NEE with restoration age**

Given the critical impact of restoration age on GHG emissions in restored ecosystems, the patterns of $CH_4$ and $N_2O$ emissions and NEE with restoration age were first explored. Overall, the restoration age had a significant effect on $CH_4$ emissions (Fig. 5c and Supplementary Fig. S3b). The *RRd* of $CH_4$ emissions in the restored forests was negatively correlated with restoration age (i.e., years since restoration) (Fig. 5c). The soil $CH_4$ uptake showed no response to the afforestation age within 10 years, while soil $CH_4$ uptake increased with afforestation age for longer time intervals (Supplementary Fig. S3b). The *RRd* of $CH_4$ emissions in the restored wetlands was exponentially and positively correlated with restoration age and achieved a relatively stable value in about 10 years since restoration (Fig. 5a). The *RRd* of $N_2O$ emissions in the wetland was negatively correlated with restoration age (Fig. 5b).

Restoration age was an important factor influencing $CO_2$ fluxes (Fig. 5). The GPP/ER showed low values (<1 or ≈1) in the early years following afforestation and restoration from fire and clear-cutting (Fig. 5g, h). The GPP/RE became greater than 1 (i.e., NEE < 0) by 4 years, 6 years, 13 years, and 8 years after restoration for the afforestation

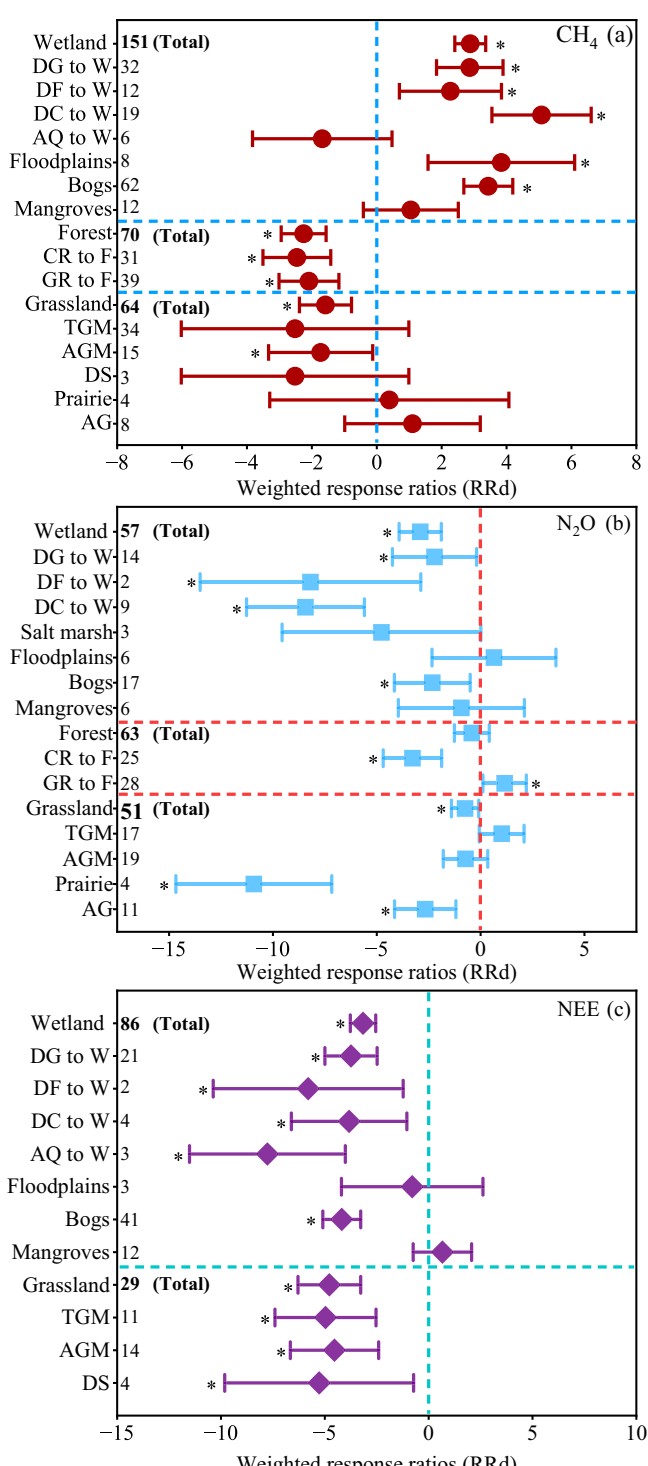

**Fig. 2 | Effects of ecological restoration on CH₄ (a), N₂O (b), and NEE (c) fluxes across the different wetland, forest and grassland restoration categories.** The overall effect size was calculated with a categorical random effects model. Values are means ±95% CIs of the weighted response ratios (*RRd*) between the paired control ecosystems and restored ecosystems. If the 95% CI value does not overlap with zero at the $\alpha = 0.05$ level, the response is considered significant. The asterisks indicate significant positive or negative effects. Numbers next to the y-axis indicate sample sizes (*n*). Due to the small sample size for the paired restored-control measurements for the NEE in forests, the effects of forest restoration on the NEE were not tested by *RRd*. DG to W, drained grassland to wetland; DF to W, drained forest to wetland; DC to wetland, drained cropland to wetland, AQ to Wetland, aquaculture to wetland; NEE net ecosystem $CO_2$ exchange, TGM temperate steppe & meadow, AGM alpine steppe & meadow, DS desert steppe, AG artificial grassland. Source data are provided as a Source Data file.

soil temperature, soil water-filled pore space (WFPS), soil moisture, and pH ($P < 0.05$; Fig. 6b). Grassland restoration remarkably increased soil WFPS, soil moisture, vegetation coverage, and grassland above-ground and belowground biomass, but reduced soil bulk density (BD) and soil $NO_3^-$-N concentrations ($P < 0.05$; Fig. 6c). Wetland restoration significantly increased water table depth, soil SOC and total N (TN), but decreased soil BD, soil redox potential (Eh), pH and $NH_4^+$-N concentrations ($P < 0.05$; Fig. 6a).

The *RRd* of CH₄ emissions in the restored forests was positively correlated with the *RRd* of soil WFPS ($P < 0.01$; Fig. 7b). The *RRd* of CH₄ emissions in the restored forests and grasslands was negatively correlated with the *RRd* of BD (Fig. 7a). Afforestation decreased CH₄ emissions regardless of the tree types (i.e., coniferous and deciduous forest) ($P < 0.01$; Supplementary Fig. S3c). The *RRd* of N₂O emissions in the forests and grasslands were positively correlated with the *RRd* of soil $NH_4^+$-N and $NO_3^-$-N concentrations (Fig. 7d, e), and the *RRd* of N₂O emissions in the forests was negatively correlated with the *RRd* of soil pH ($P < 0.01$; Fig. 7f).

The *RRd* of CH₄ emissions in the restored wetlands was exponentially and positively correlated with water table depth ($P < 0.01$; Fig. 5a and Fig. 7c). The *RRd* of N₂O emissions in the wetland was positively correlated with the *RRd* of soil $NH_4^+$-N concentrations (Fig. 7g). The N₂O emissions and NEE of the restored wetlands were negatively related to water table depth ($P < 0.001$; Supplementary Fig. S4c).

Across all restoration groups, GPP and ER were positively correlated with the temperature and precipitation ($P < 0.01$; Supplementary Fig. S5). The *RRd* of CH₄ emissions in all systems was negatively correlated with the aridity index ($P < 0.01$; Supplementary Fig. S6c). When the precipitation was larger than 900 mm, the *RRd* of N₂O emissions in all ecosystems was positively correlated with precipitation ($P < 0.05$; Supplementary Fig. S6e). The *RRd* of NEE in all ecosystems was negatively correlated with the aridity index when the aridity index was greater than 0.9 ($P < 0.05$; Supplementary Fig. S6i).

## Discussion

### Forest and grassland restoration increased CH₄ uptake while wetland restoration enhanced CH₄ emissions

We found that forest and grassland restoration significantly increased CH₄ uptake (Table 1, Fig. 2), suggesting the great potential of forest and grassland ecosystem restoration in enhancing sink function for CH₄. The conversion of croplands and grasslands to forests increased CH₄ uptake by 84.8% and 106.8% (Fig. 3), respectively, indicating high CH₄ removal efficiency. The *RRd* of CH₄ emissions linearly increased with the *RRd* of WFPS (Fig. 7b), indicating that lower soil moisture and WFPS enhanced CH₄ uptake and inhibited CH₄ emissions from soil. Afforestation significantly decreased WFPS in the forest ecosystems (Fig. 6b), mainly because trees had deeper roots and higher water demands than crops and grasses, and afforestation enhanced evapotranspiration and canopy interception of precipitation[39,40]. The

sites, clear-cutting sites, post-fire sites, and all disturbances sites, respectively (Fig. 5). The GPP/ER ratio varied with afforestation age and time since restoration after disturbance, and had an asymptote of 1.19 and 1.09, respectively (Fig. 5g, h; $P < 0.001$). The NEE in the wetlands was exponentially and negatively correlated with restoration age and the switchover time from net $CO_2$ sources to net $CO_2$ sinks was estimated to be approximately 4 years (Fig. 5d).

### Factors influencing the response of CH₄, N₂O, and NEE to ecological restoration

Forest restoration significantly increased concentrations of soil organic C (SOC), $NH_4^+$-N and dissolved organic C (DOC), but reduced

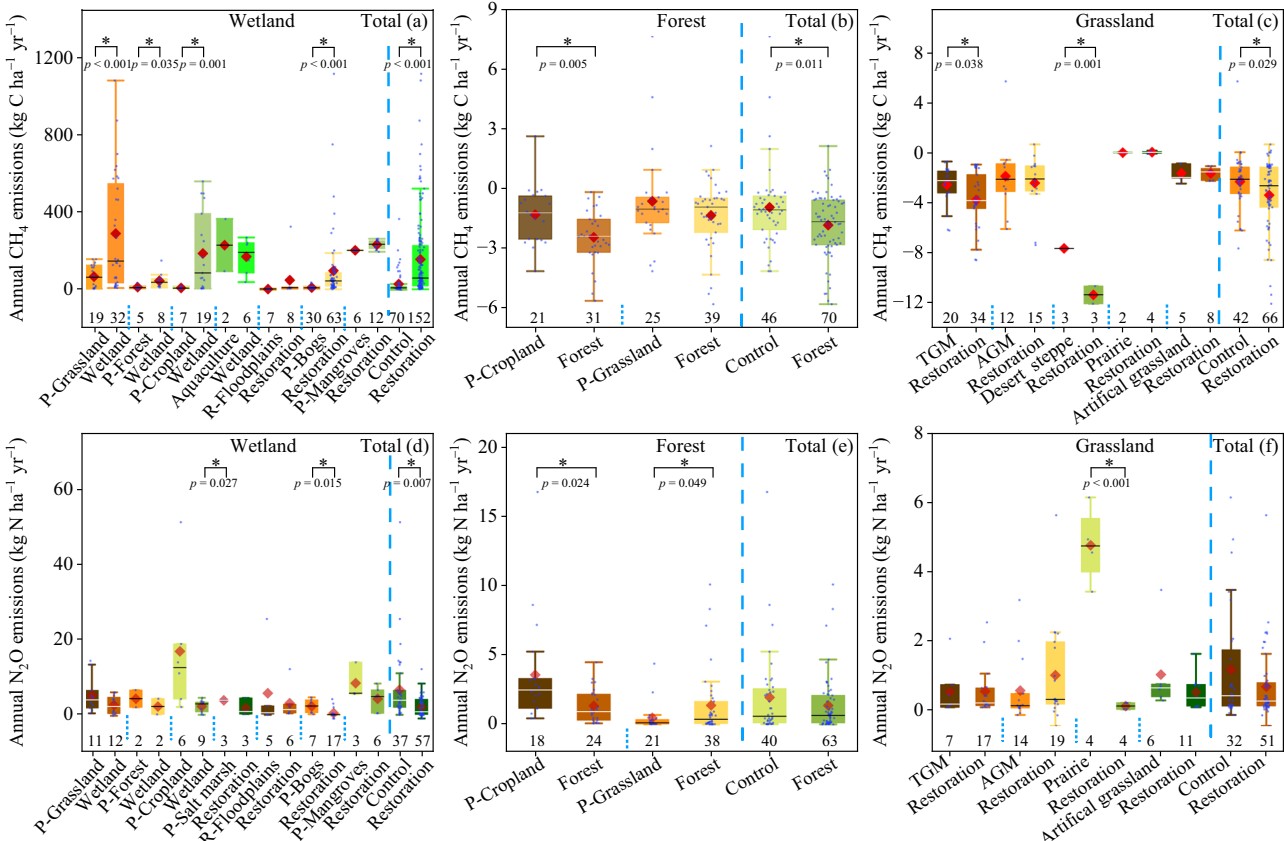

**Fig. 3 | Box plots of CH₄ and N₂O fluxes in the restored wetlands and their paired control ecosystems (a and d), the restored forests and their paired control ecosystems (b and e), the restored grasslands and their paired control ecosystems (c and f).** Every two adjacent boxes represent the paired control-restored measurements. The paired control ecosystems are prefixed with 'P'. TGM, temperate steppe & meadow; AGM, alpine steppe & meadow. Box boundaries represent the 75th and 25th percentiles, whisker caps represent the 95th and 5th percentiles, and circle points represent outliers. Diamond points and solid lines inside the boxes represent means and medians, respectively. Asterisks (*) denote significance at $p < 0.05$, as determined by using a two-sided, independent samples $t$ test. No adjustments were made for multiple comparisons. Numbers next to the x-axis indicate sample sizes ($n$). Exact $p$-values and Source data are provided as a Source Data file.

decrease in soil WFPS caused by afforestation can enhance the diffusion of atmospheric $O_2$ and $CH_4$ into the soils, thereby increasing $CH_4$ oxidation and uptake in the afforested soils[39]. Soil compaction by machinery in the agricultural lands and trampling by livestock in the grasslands may increase soil bulk density and reduce soil porosity[41–43]. Our results showed that grassland restoration significantly reduced soil bulk density (Fig. 6c), and the *RRd* of $CH_4$ emissions showed a negative relationship with the *RRd* of soil bulk density (Fig. 7a), implying that the lower soil bulk density in the restored grasslands increased $CH_4$ diffusion from atmosphere into soils and thus increased $CH_4$ uptake[44]. Grassland restoration by reducing grazing intensity or grazing exclusion increased belowground biomass (Fig. 6c), which may form "root holes" and improve soil aerobic conditions for diffusion of atmospheric $CH_4$ into the soil profiles and the growth of methanotrophs[45], thereby enhancing $CH_4$ uptake in the restored grassland. In addition, the increase in SOC in the afforested lands (Fig. 6b) could enhance soil macropores and the number of coarse pores[46], and thus create favorable environments for methanotrophs growth and $CH_4$ oxidation[47]. Taken together, the increased $CH_4$ uptake in the restored forests and grasslands could be attributed to the changes in soil properties.

Forest restoration significantly increased $CH_4$ uptake with the increase of afforestation age (Fig. 5c and Supplementary Fig. S3b), which could be mainly attributed to the increased SOC and decreased soil moisture and WFPS following afforestation (Figs. 6b and 7b)[48,49]. Bárcena et al. reported that soil SOC concentrations increased with stand age and therefore increased the abundance and activity of methane-oxidizing bacteria growth by supplying abundant substrates[50], consequently resulting in an enhanced $CH_4$ oxidation rate with afforestation age. Gatica et al. found that, with the increase of afforestation age, soil moisture was decreased by the combined effects of increasing rainfall interception[51] and tree transpiration in the older forest stands[39], and thus enhanced $CH_4$ consumption with time. Therefore, these individual observations support our results and inferences[48–51].

We found that wetland restoration significantly increased annual $CH_4$ emissions by 5.4 times compared with the paired control ecosystems, indicating that wetland restoration enhanced the $CH_4$ source strength. Among the types of wetland restoration, the conversion of grasslands to wetlands showed the largest increase in $CH_4$ emissions, followed by the conversion of croplands to wetlands (Fig. 3a). These results indicated that greater attention should be paid to the increased $CH_4$ emissions from the restored wetlands in global GHG accounting. Previous work showed that $CH_4$ emissions in the wetlands were mainly controlled by water table level[26,38], nutrient status[44], plant species[52], and microbial activity[53]. Unlike the $CH_4$ emissions in forest and grassland ecosystems which were influenced by soil properties (Fig. 7a, b), the *RRd* of wetland $CH_4$ emissions was positively correlated with water table depth, suggesting that the wetland restoration facilitated the production of $CH_4$ by creating an anaerobic environment through elevated water table levels (Figs. 6a, 7c). Long-time waterlogging during the restoration period reduced $O_2$ penetration into the sediments

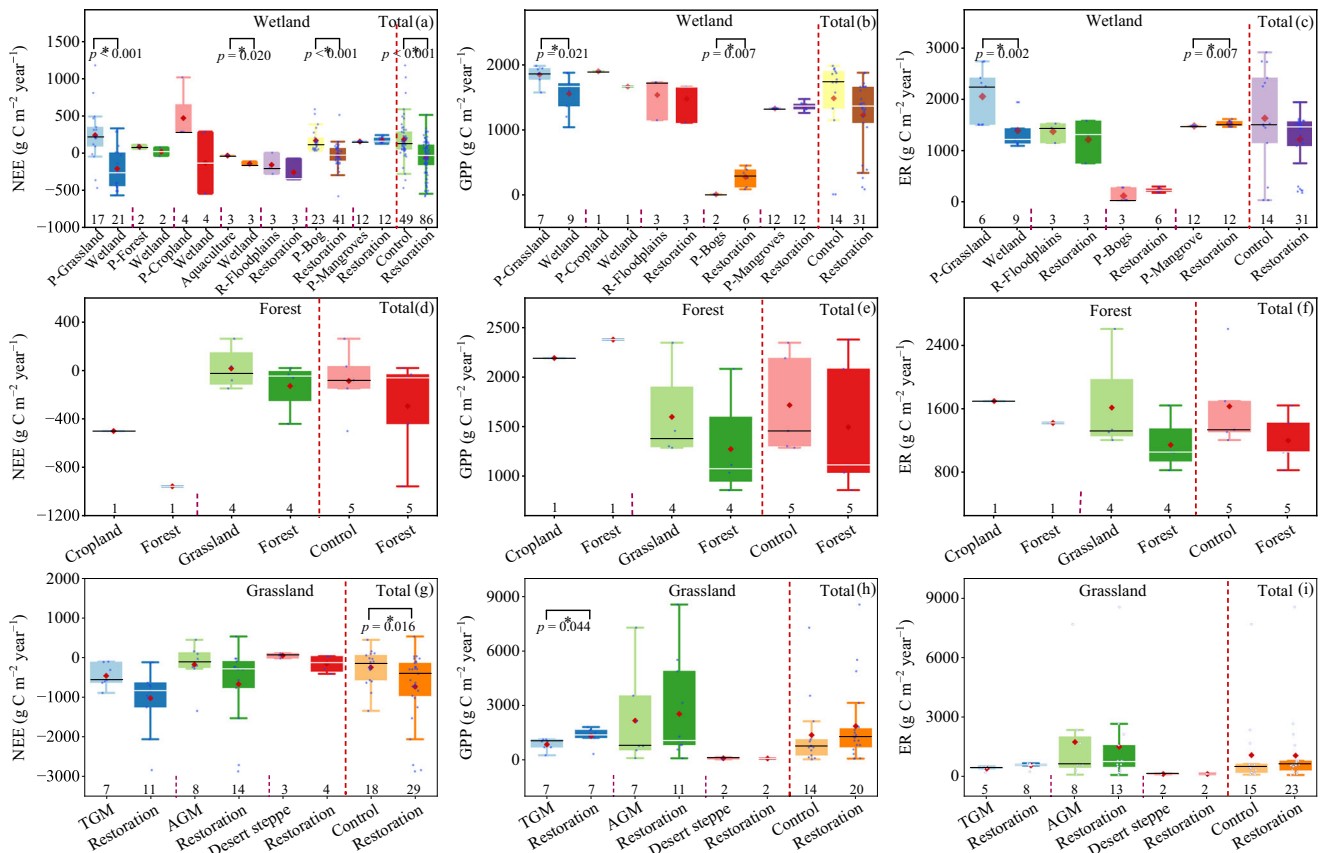

**Fig. 4 | Box plots of annual $CO_2$ fluxes (NEE, GPP and ER) the restored wetlands and their paired control ecosystems (a–c), the restored forests and their paired control ecosystems (d–f), and the restored grasslands and their paired control ecosystems (g–i).** Every two adjacent boxes represent the paired control-restored measurements. The paired control ecosystems are prefixed with 'P'. Box boundaries represent the 75th and 25th percentiles, whisker caps represent the 95th and 5th percentiles, and circle points represent outliers. Diamond points and solid lines inside the boxes represent means and medians, respectively. Asterisks (*) denote significance at $P < 0.05$, as determined by using a two-sided, independent samples $t$ test. No adjustments were made for multiple comparisons. Numbers next to the x-axis indicate sample sizes (n). NEE net ecosystem $CO_2$ exchange, GPP, gross primary productivity, ER ecosystem respiration, TGM temperate steppe & meadow, AGM alpine steppe & meadow. Due to the small sample size for the paired restored-control measurements for the NEE, GPP, and ER in forests, the effects of forest restoration on $CO_2$ fluxes were not tested by the t-test. Exact $p$-values and Source data are provided as a Source Data file.

**Table 1 | Changes in comprehensive C budget and GWP when converting the paired control ecosystems to the restored ecosystems (Mean ± SE)**

| Ecosystem | Restoration type | $CH_4$ kg C ha⁻¹ year⁻¹ | $N_2O$ kg N ha⁻¹ year⁻¹ | NEE g C m⁻² year⁻¹ | C budget g C m⁻² year⁻¹ | GWP t $CO_2$-eq ha⁻¹ year⁻¹ | Rate of change % |
|---|---|---|---|---|---|---|---|
| Forest | Control | −1.0 ± 0.3 | 2.0 ± 0.5 | −87.7 ± 124.7 | −87.8 ± 124.8 | −2.4 ± 4.8 | |
| | Restoration | −1.9 ± 0.2 | 1.4 ± 0.3 | −295.2 ± 184.7 | −295.5 ± 184.7 | −10.3 ± 6.9 | −327.7 |
| Grassland | Control | −2.6 ± 0.3 | 1.2 ± 0.3 | −205.0 ± 64.9 | −205.2 ± 65.0 | −7.1 ± 2.5 | |
| | Restoration | −3.4 ± 0.3 | 0.9 ± 0.2 | −506.2 ± 113.4 | −506.5 ± 113.5 | −18.3 ± 4.2 | −157.7 |
| Wetland | Control | 23.4 ± 6.9 | 6.7 ± 1.6 | 176.5 ± 42.1 | 178.8 ± 42.7 | 10.2 ± 2.5 | |
| | Restoration | 150.8 ± 17.1 | 2.1 ± 0.3 | −68.5 ± 25.6 | −53.4 ± 27.3 | 3.9 ± 1.7 | −62.0 |

*NEE* net ecosystem $CO_2$ exchange, *GWP* global warming potentials, *C budget* the sum of NEE-C and $CH_4$-C. Source data are provided as a Source Data file.

and thus induced a reduction in the redox potential of 1.23-fold compared with the paired control ecosystems (Fig. 6a), which in turn stimulated methanogen growth and activity, thereby enhancing $CH_4$ emissions[54]. Bog restoration by rewetting may be beneficial to the proliferation of aerenchymatous vascular plants, and thus allow $CH_4$ to bypass the oxidized surface soil, consequently enhancing $CH_4$ emission into the atmosphere via the plant-mediated transport[32]. In addition, restoration of wetlands by rewetting created an anaerobic environment which may inhibit microbial activity and reduce SOC decomposition[55], thus leading to a higher SOC concentration in the

restored wetlands (Fig. 6a). The higher SOC concentrations in the restored wetlands could provide more substrates for methanogen growth than in the paired control ecosystems[44,56]. Thus, elevating the water table level and increasing the substrate supply for methanogen is likely to result in higher $CH_4$ fluxes in the restored wetlands (Figs. 6a, 7c, and Supplementary Fig. S4a).

The *RRd* of wetland $CH_4$ showed exponentially relationship with restoration age and achieved a relatively stable value in about 10 years since restoration (Fig. 5a). Similarly, Mitsch et al. reported that wetland restoration initially stimulated $CH_4$ emissions but decreased over time

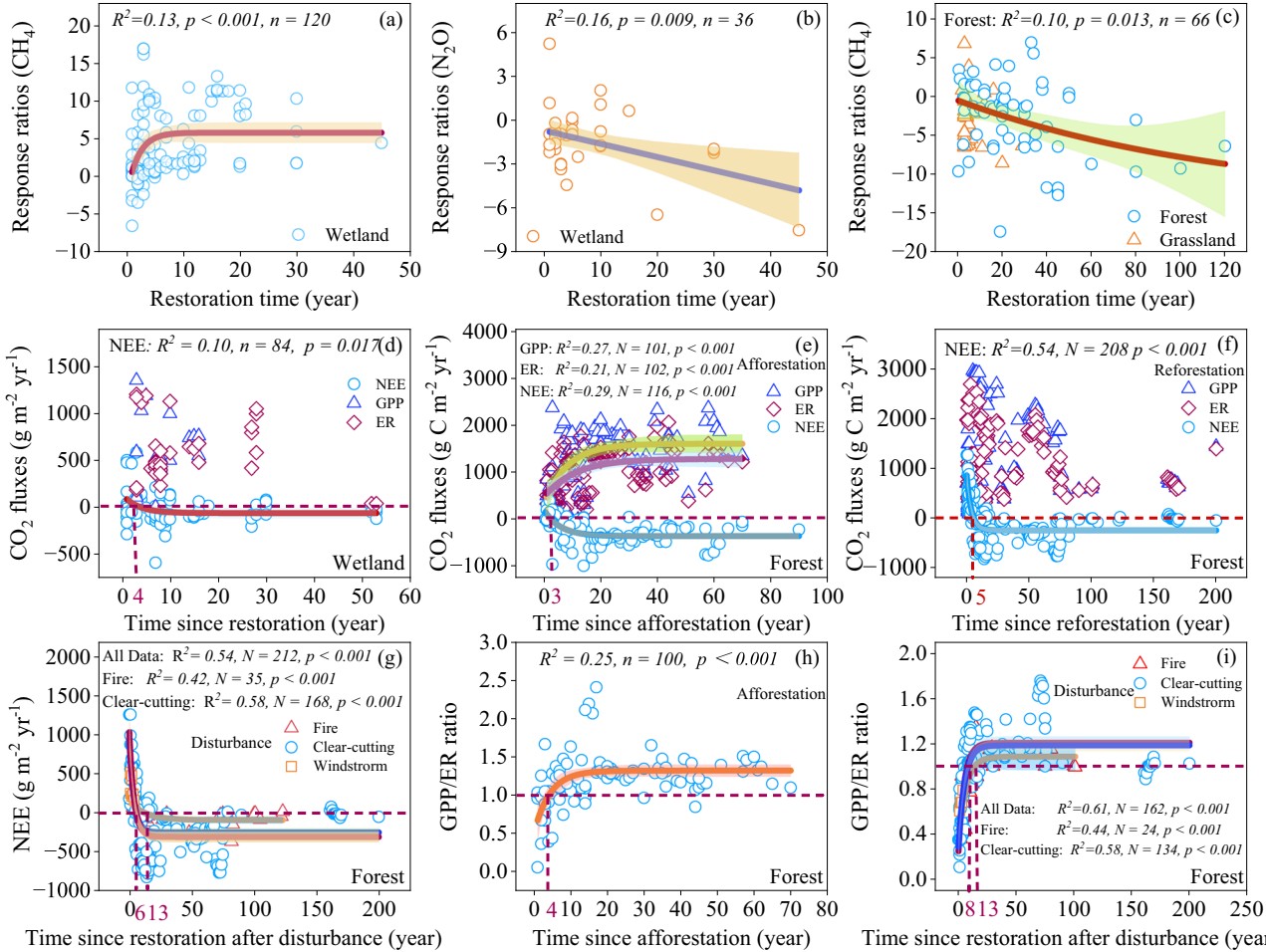

**Fig. 5 | Relationships of the response ratios (*RRd*) of CH$_4$ and N$_2$O fluxes, and the annual CO$_2$ fluxes with restoration age. a, b** relationships between the response ratios of wetland CH$_4$ (**a**) and N$_2$O (**b**) fluxes and restoration age. **c** Relationship between the response ratios of forest and grassland CH$_4$ fluxes and restoration age. **d–f** relationships between the CO$_2$ fluxes in wetland (**d**) and forest (**e** and **f**) and restoration age. **g** relationships between the fluxes of NEE and restoration age.

**h**, **i** relationships between GPP/ER and afforestation time (**h**), and restoration age after forest disturbance (**i**). Linear and nonlinear regression were used and the error bands surrounding the regression lines represent the 95% confidence interval of the correlation. The n is the number of observations. Exact *p*-values and Source data are provided as a Source Data file.

and reached CH$_4$ emissions comparable to the natural wetland after 13-15 years[28]. The rapid response of CH$_4$ emissions to wetland restoration at the initial stage was mainly due to the restoration of the natural hydrology and the inundation of easily decomposable plant litters, which created an anaerobic environment and sufficient substrate for the growth of methanogens and CH$_4$ production[53,57–59].

**Diverse responses of N$_2$O emissions to ecological restoration**
Our findings revealed that the conversion of agricultural lands to forests significantly decreased N$_2$O emissions and the conversion of grasslands and wetlands to forests stimulated N$_2$O emissions, indicating the response patterns of N$_2$O emissions to forest restoration depends on the prior land-use type. The *RRd* of N$_2$O emissions in forests was positively related to the *RRd* of NH$_4^+$ and NO$_3^-$ (Fig. 7), indicating that the cessation of fertilization in the afforested croplands may lead to a reduction in soil N$_2$O emissions compared with the fertilized croplands[11]. The main reasons for the increased N$_2$O emissions in the forests converted from grasslands could be attributed to that afforestation in grasslands increased the concentrations of soil SOC and NH$_4^+$ and decreased soil pH (Fig. 7, Supplementary Fig. S7). The increased SOC and NH$_4^+$ could increase soil C and N availability and soil nitrification for N$_2$O production[12,60]. Our results demonstrated that the *RRd* of N$_2$O emissions showed a negative relationship with the *RRd* of

soil pH (Figs. 6b, 7f). The reduction of soil pH may inhibit the activity of the N$_2$O reductase enzyme and in turn increase N$_2$O/N$_2$ ratios in the denitrification, consequently increasing N$_2$O emissions from denitrification in the afforested soils[4,61]. Grassland restoration by conversion of cropland to grassland sharply decreased N$_2$O emissions (Supplementary Fig. S1b), mainly due to the decreased concentrations of soil NO$_3^-$ by stopping fertilization (Figs. 6c, 7). In contrast, artificial assisted restoration in the degraded grasslands by applying organic or mineral fertilizer may increase N$_2$O emissions by increasing the availability of N for N$_2$O production[62]. Thus, our results indicated that different grassland restoration measures showed distinct impacts on N$_2$O emissions (Supplementary Fig. S1b).

Wetland restoration significantly decreased N$_2$O emissions (by 68.6%) and soil NH$_4^+$ concentration, and the *RRd* of N$_2$O emissions in wetlands was positively related to the *RRd* of NH$_4^+$ (Figs. 2b, 3d, 6a, 7g), indicating that the reduction in soil NH$_4^+$ concentration, driven by wetland restoration, contributes to the decreased N$_2$O emissions in the restored wetlands (Fig. 6a). Previous work showed that the conversion of agricultural lands and grasslands to wetlands significantly decreased N fertilizer and animal waste inputs, thus reducing the substrates of inorganic N for nitrifying and denitrifying microorganisms[63]. Raising the water table in rewetted peatlands can potentially increase the diffusional barrier for deep soil N$_2$O

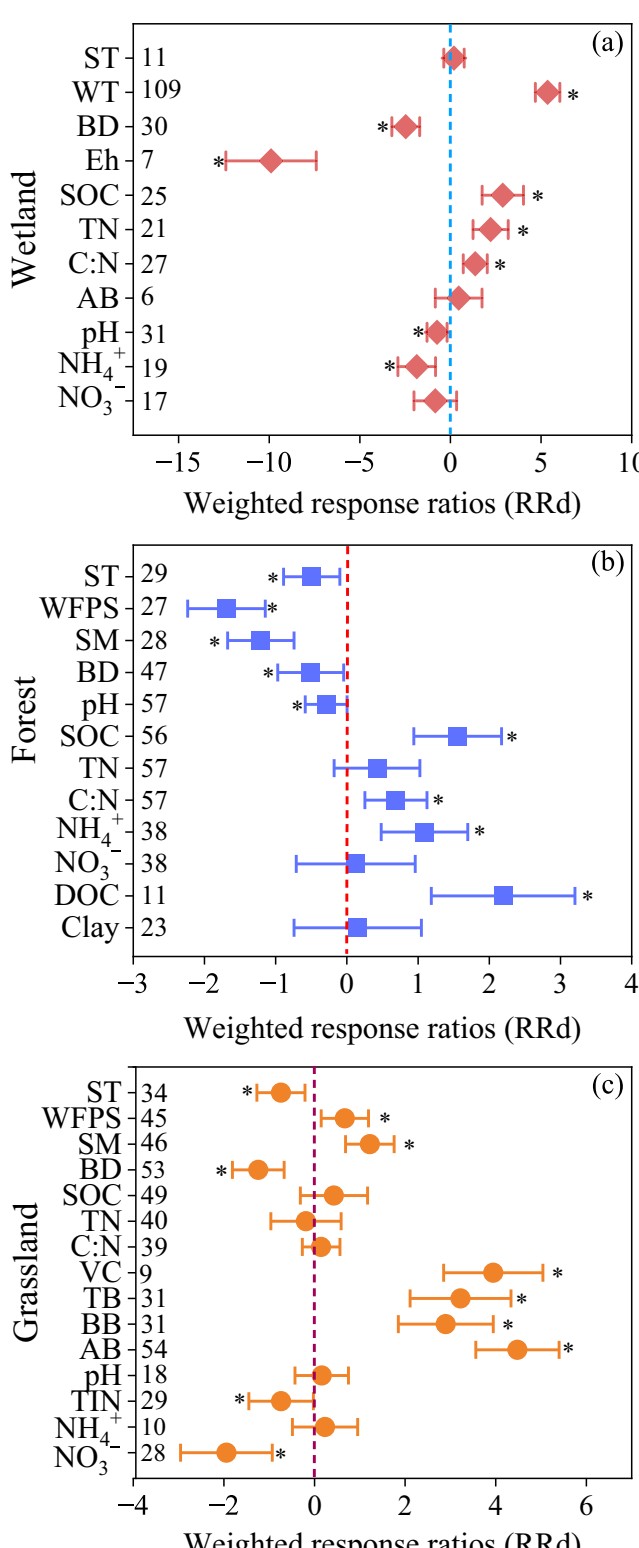

**Fig. 6 | Effects of ecological restoration on soil properties in wetland (a), forest (b), and grassland (c) ecosystems.** The overall effect size was calculated with a categorical random effects model. Values are meant ±95% CIs of the weighted response ratios (*RRd*) between the paired control ecosystems and restored ecosystems. If the 95% CI value does not overlap with zero, the response is considered significant. The asterisks indicate significant positive or negative effects. SOC soil organic carbon, TN total N, C: N carbon/nitrogen ratio, Eh soil redox potential, BD bulk density, SM soil moisture, WFPS water-filled pore space, ST soil temperature, WT water table depth, VC vegetation coverage, TB total biomass, AB aboveground biomass, BB belowground biomass; $NH_4^+$ ammonium, $NO_3^-$ nitrate. Source data are provided as a Source Data file.

### Forest, grassland, and wetland restoration enhances C sink and reduces the GWP

Our results demonstrated that the NEE decreased with afforestation and reforestation age, and the estimated time required for the transition from $CO_2$ sources to net sinks was approximately 3-5 years (Fig. 5), indicating that restored forests have the capacity to rapidly become $CO_2$ sinks. At the early stage of restoration, forests may act as weak $CO_2$ sources (Fig. 5), primarily due to the low foliar biomass and the rapid decomposition of residuals in the ground and soils[17]. As trees grow, the increases in GPP surpassed the rise in ER. The enhanced ability of C assimilation and the subsequent increase in annual woody biomass production are the key factors driving restored forests to function as $CO_2$ sinks[67-69]. The estimated switchover time of restored forests from $CO_2$ source to net sink after the disturbance was approximately 6 years for the clear-cutting sites and 13 years for the post-fire sites (Fig. 5), indicating a relatively slow recovery of the C sink function in the burned sites. Forest restoration progressively increased the GPP/ER ratio with restoration age, eventually reaching a stable value of 1.1-1.2 after approximately 20 years. Interestingly, this value aligns with the average GPP/ER ratio of 1.2 observed in mature forests worldwide[70].

Grassland restoration markedly reduced the NEE (Fig. 2c), suggesting that grassland restoration effectively increased C sink capacity. The *RRd* of NEE in grassland was positively related to the *RRd* of soil moisture (Fig. 7h), indicating that higher soil moisture increased grassland $CO_2$ sinks. Grassland restoration by grazing exclusion or reducing grazing density is conducive to the recovery of grassland and the increase of vegetation coverage (Fig. 6c), thereby reducing evaporative water loss from the soil[71]. The increase in soil moisture in the restored grassland decreased NEE by increasing GPP relatively more than ER[72]. The results are consistent with the general observations that higher soil water content increased vegetation leaf area index and GPP, and thereby resulted in a great C sink capacity in the grassland ecosystems[73]. In addition, the significant relationship between grassland NEE and aboveground biomass suggested that grassland restoration by grazing exclusion favored the regrowth of grasses and increased above- and belowground biomass (Figs. 6c and 7h), and thus increased gross ecosystem photosynthesis and eventually resulted in a significant decrease in NEE[74]. Grassland restoration by converting cropland to grassland remarkably decreased NEE (Supplementary Fig. S1c), indicating an increased C sink. The large residual of dead roots in the soil with cropland harvest would decompose and release large $CO_2$ emissions, while the grassland had a large live root and thereby decreased ecosystem respiration and increased net $CO_2$ uptake in the grassland systems[75]. Therefore, the enhanced $CO_2$ uptake in the restored grassland was mainly due to the increase in soil moisture and vegetation biomass (Figs. 6c, 7h).

Overall, wetland restoration significantly reduced the NEE and shifted the ecosystems into $CO_2$ sinks (Fig. 4a), highlighting the effectiveness of wetland restoration in enhancing $CO_2$ sequestration. The significant negative correlation of NEE with water table depth in the restored wetlands indicated that the rise of the water table plays a crucial role in promoting the $CO_2$ sink (Supplementary Fig. S4c). This

emissions into the atmosphere and enhance the microbial complete reduction of $N_2O$ to $N_2$ by denitrifying bacteria[30,64,65], thereby reducing $N_2O$ emissions in the restored peatlands[30]. In addition, Brummell et al. attributed the reduction of $N_2O$ emissions in the restored peatlands to the rapid reestablishment of vascular plants[66], which effectively take up soil N by plant roots and thereby decrease N availability for nitrification and denitrification. Therefore, reducing soil N availability and raising water levels could create unfavorable environments for $N_2O$ emissions.

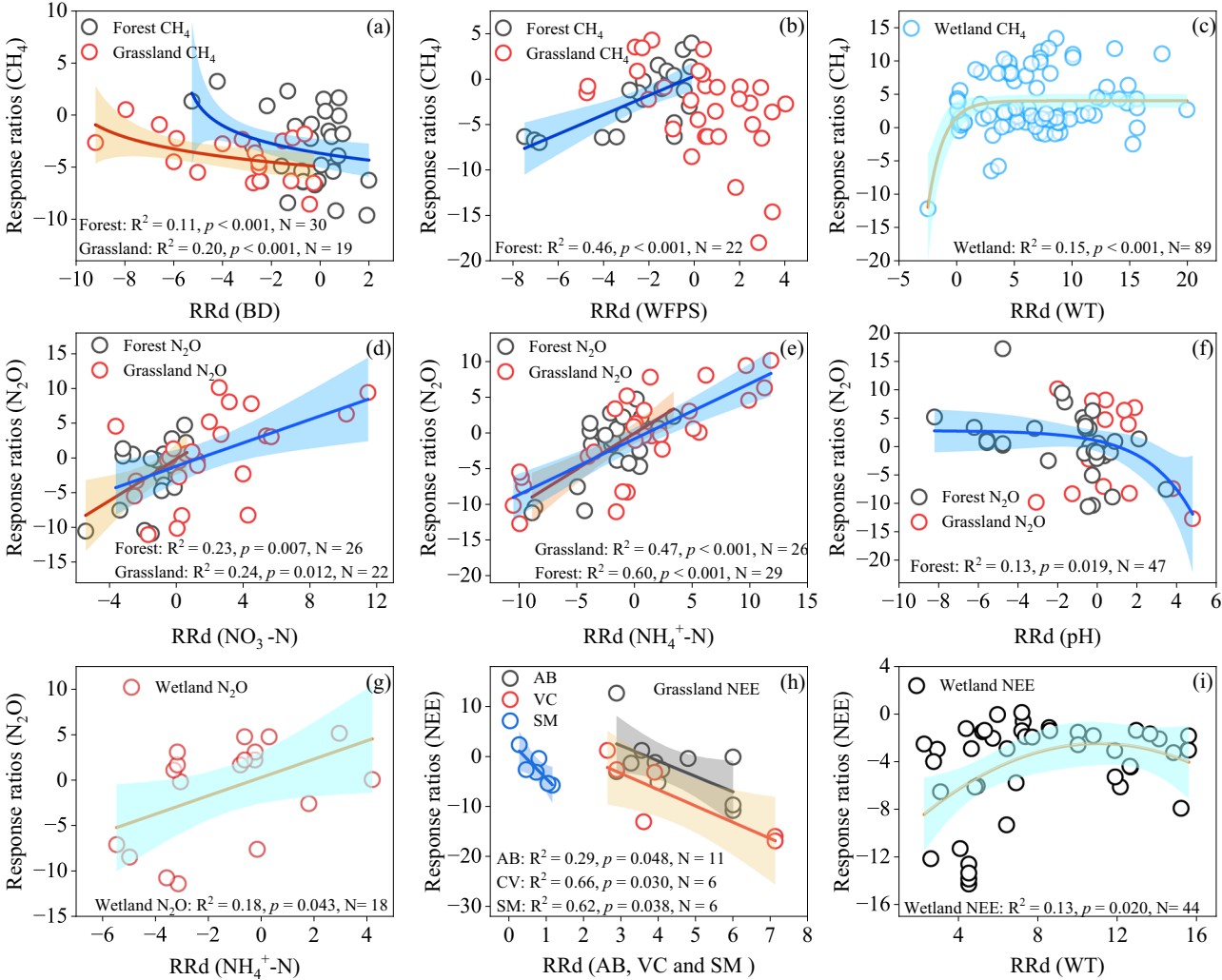

**Fig. 7 | Relationships of the response ratios (*RRd*) of CH$_4$ (a–c), N$_2$O (d–g), and NEE (h and i) with response ratios (*RRd*) of soil properties after wetlands, forest, and grassland ecosystems were restored.** WT water table depth, BD bulk density, SM soil moisture, WFPS water-filled pore space, AB aboveground biomass, VC vegetation coverage. Linear and nonlinear regression were used and the error bands surrounding the regression lines represent the 95% confidence interval of the correlation. The n is the number of paired observations. Exact *p*-values and Source data are provided as a Source Data file.

pattern is expected since individual observations have confirmed that restoring the drained bog by raising the water table could create suitable conditions for vegetation growth, and the recolonization of bryophytes and vascular plants in restored wetlands contributed to the net $CO_2$ uptake[26,27,29]. Returning aquaculture to wetland significantly enhanced $CO_2$ sinks (Fig. 2c), which may be mainly due to the increase of the aquatic vegetation biomass and the shift of dominant species from submerged vegetation to flotation vegetation[76]. In addition, we found that the NEE of wetlands showed a negative exponential relationship with restoration age, and the transition time from net $CO_2$ sources to net $CO_2$ sinks was estimated to be around 4 years (Fig. 5d). Similarly, Waddington et al. reported that the restored peatlands could serve as net C sinks (20 g C m$^{-2}$) after two years restoration, mainly due to the rapid increase in moss cover and biomass after rewetting[77]. Lee et al. reported that the eighth year of the restored peatlands exhibited a net C sink (163 g C m$^{-2}$ yr$^{-1}$)[78], falling within the range of C sink values reported for pristine peatlands (50-267 g C m$^{-2}$ yr$^{-1}$)[26,79–81]. Over longer time frames, we can anticipate a gradual stabilization of the $CO_2$ sink as the biomass pool in the restored wetland approaches a steady C sequestration rate.

Regarding the C balance (excluding DOC fluxes), wetland restoration increased CH$_4$ emissions and $CO_2$ uptake, resulting in net sinks with a mean of 53 g C m$^{-2}$ yr$^{-1}$ (Table 1). The balance between $CO_2$ sinks and the strong warming potentials of CH$_4$ and N$_2$O emissions strongly determined the net climatic impact of the restored wetlands[78]. We found that wetland restoration resulted in a significant reduction in the GWP (Table 1), mainly due to the substantial reduction in $CO_2$ emissions and the accompanying decrease of N$_2$O emissions compared with the paired control ecosystems. Generally, wetland restoration could reduce GWP by 43-90% (Table S1). Among the wetland restoration measures, rewetting, moss layer transfer, and replanting & rewetting could significantly decrease GWP (Table S2). Returning croplands to wetlands is the most effective way to reduce GWP (Table S1). Our results were in accordance with the previous findings that wetland restoration remarkably decreased GWP to a neutral range (3.3 to 6.7 t $CO_2$-eq ha$^{-1}$ yr$^{-1}$) at the restored wetlands in comparison with the drained wetlands, mainly due to the increased $CO_2$ uptake by precluding oxidation of the residual in the restored wetlands[29,33,78]. Previous work found that the restoration of bogs through rewetting could yield exceptionally high $CO_2$ sink, which could effectively offset CH$_4$ emissions, thus resulting in a substantial reduction in the GWP[29]. Afforestation increased CH$_4$ and $CO_2$ uptake (Table 1), thus leading to a significant reduction in GWP with a mean of −10.3 t $CO_2$-eq ha$^{-1}$ yr$^{-1}$. Afforestation provided an effective strategy for

GWP mitigation largely due to the increase in woody biomass C[82]. Grassland restoration resulted in a decrease in GWP across all the grassland types and restoration measures (Table 1, Table S2 and S3), mainly by increasing the net $CO_2$ sink. Similarly, Rong et al. observed that grassland restoration brought about a greater increase in GPP than ER in a heavily grazed grassland, therefore decreasing NEE in the grassland[72]. Taking together, our study suggested that, forest, grassland, and wetland restoration could serve as an effective strategy for mitigating GHG emissions and reducing GWP.

**Implications for the IPCC reports, Guidelines for National Greenhouse Gas Inventories, and future research**

The present study provides a comprehensive understanding of the impacts of ecosystem restoration on GHG emissions at a global scale. Based on the compiled dataset, our results demonstrate that ecological restoration has vast potential to mitigate GHG at a global scale and provide insights and data for the IPCC reports and Guidelines for National Greenhouse Gas Inventories, particularly in relation to ecological restoration and land use change. In the context of "Agriculture, Forestry and Other Land Use (AFOLU)", wetlands are recognized for their high level of uncertainty in the national greenhouse gas inventory reports submitted by States Parties to the United Nations Framework Convention on Climate Change (UNFCCC). Over the years, the IPCC has issued a series of methodological documents, such as "IPCC 2006 Guidelines for National Greenhouse Gas Inventories", "2013 Supplement to the 2006 IPCC Guidelines for National Greenhouse Gas Inventories", and "2019 Refinement to the 2006 IPCC Guidelines". However, there is very little data regarding ecological restoration and its impacts on GHG emissions. The emission factors provided by the IPCC mainly focus on the dynamics of GHG in drained and rewetted organic soils[83–85]. Our study has expanded the available datasets for the restored wetlands and their paired control lands including different wetland types (bogs, mangroves, and floodplains wetland) and various land-use changes (i.e., conversion of cropland to wetland, conversion of grassland to wetland, conversion of forest to wetland, and returning aquaculture to wetland) (Figs. 3, 4). In addition, our study provides an updated $N_2O$ emission factor for rewetted wetlands, estimated at 2.1 kg N ha$^{-1}$ yr$^{-1}$ (95% CI: 1.4 to 2.8) (Fig. 3d), which is considerably higher than the assumed default value of 0 kg N ha$^{-1}$ yr$^{-1}$ for rewetted organic soils in the IPCC guidelines due to the limited data[79]. For the forests, the IPCC Guidelines only considered non-$CO_2$ gases from biomass burning and assumed that the conversion of croplands, grasslands and other lands to forest lands tended not to alter the sources and removals of non-$CO_2$ gases[85]. However, this assumption may not always hold true mainly due to the changes in soil properties resulting from land conversion (Fig. 6 and Supplementary Fig. S7)[12,21]. In our study, we conducted a detailed analysis for each land conversion type individually (i.e., conversion of croplands to forests and conversion of grasslands to forests) and developed a meta-data for non-$CO_2$ gases (i.e., $CH_4$ and $N_2O$ emissions) (Figs. 2, 3), thereby providing valuable data for refining the non-$CO_2$ gases inventory methodology for the conversion of other lands into forests.

Our results confirmed that afforestation and reforestation, as well as rewetting the drained wetlands, should become critical for future ecological restoration to mitigate GHG (Tables S1 and S2, Fig. 5). In addition, previous work reported that the aquaculture systems are an important source of $CH_4$[86], and our results verified that returning aquaculture to wetlands was an effective measure to enhance C sinks (Table S1). Wetland restoration by converting drained forests, grasslands, and croplands into wetlands increased $CO_2$ sinks by decreasing ecosystem respiration, effectively transforming the ecosystems from $CO_2$ sources to sinks[29]. Bog restoration by rewetting can be an active restoration strategy to recover vegetation and convert the extracted bog from GHG sources to GHG sinks[26] (Table S2). For the grassland,

restoring the degraded grassland either by grazing exclusion, reducing grazing intensity, or converting croplands to grassland is an effective strategy for mitigating GHG (Tables S2 and S3). These findings offer valuable insights for policymakers to select effective ecological restoration measures.

Our study highlights the significance of restoration age in regulating GHG emissions in restored ecosystems, underscoring the importance of considering the time in assessing or modeling the effects of restoration or land-use change on GHG emissions (Fig. 5 and Supplementary Fig. S3). Although restoration measures can be implemented and completed quickly, the re-establishment of plant coverage and microbial communities is a gradual process[50,77,78]. The process of biomass accumulation changes over time[15,77], and soil physical, biogeochemical, and hydrological properties change with restoration time[39,41,50]. Consequently, restoration age plays a significant role in regulating GHG budgets. Although biogeochemical processes in restored ecosystems have been studied in recent years[12,16,30,39], there remains considerable uncertainty regarding the duration required for a restored ecosystem to transition into a net $CO_2$ sink. By compiling data from peer-reviewed literature, we identified the temporal patterns of NEE for forest and wetland restorations and determined the switchover time needed for the restored ecosystem to become a $CO_2$ sink. These temporal patterns of NEE highlighted the need for policymakers and planners to prioritize measures that facilitate the long-term recovery of the degraded systems in order to maximize climatic benefit and better achieve the goals of the UN Decade on Ecosystem Restoration (2021-2030). Moreover, the empirical equations and insights gained from our study regarding temporal patterns of NEE following restorations can provide important information for ecological modeling efforts. In addition, the soil $CH_4$ and $N_2O$ emissions are mainly governed by methanogens, methanotrophs, nitrifying and denitrifying microbes. However, studies about the effects of ecological restoration on these microbial communities are still insufficient[57,87], which limits the explanation and prediction of the patterns of $CH_4$ and $N_2O$ emissions under ecological restoration. Therefore, future investigations should prioritize examining the microbial mechanisms underlying changes in soil $CH_4$ and $N_2O$ emissions during the ecological restoration process.

## Methods
### Data source
We compiled a global dataset on GHG emissions associated with ecological restoration from published literature (Fig. 1). We systematically searched the peer-reviewed literature from Google Scholar, Web of Science, and the China National Knowledge Infrastructure using the following keywords: TS = (restoration * OR rehabilitation * OR revegetation * OR recovery * OR reconstruction * OR reclamation * OR restored * OR restoring * OR recovering *) AND TS = (methane * OR $CH_4$ * or nitrous oxide * OR $N_2O$ * or carbon dioxide * OR $CO_2$ * or greenhouse gas *) AND TS = (wetland * or forest * or grassland *). The search results were filtered to include articles published between December 1999 and June 2023. Peer-reviewed studies were selected by the following criteria: (1) the selected experiments were conducted in the field from restored sites with paired control sites, or chronosequence sites; (2) each treatment was required to have at least three replicates; (3) the measurement covered an entire year or at least one growing season[88]; (4) the selected studies reported at least one type of GHG. Finally, the dataset used in this study included a paired restored-control samples sub-dataset and a chronosequence sub-dataset, which were compiled from 253 peer-reviewed articles (Supplementary Fig. S8). The paired sub-dataset included 679 paired measured cases, and the chronosequence sub-dataset included 1289 data points with restoration age (i.e., years since restoration) (Supplementary Data 1–5).

The dataset included: (1) GHG fluxes, including $CH_4$, $N_2O$, GPP, ER and NEE; (2) environmental factors, including longitude, latitude, mean annual air temperature (MAT), and mean annual precipitation (MAP); (3) restoration age, i.e., the years since restoration; (4) soil properties obtained from individual studies, including soil water table depth (WT), soil temperature (ST), and WFPS, Eh, BD, soil pH, SOC, TN, soil $NH_4^+$ and soil $NO_3^-$. The fluxes of $CH_4$, $N_2O$ and $CO_2$ were usually measured with the static chamber technique and eddy covariance method. The NEE is calculated as the differences of GPP and RE. If the GPP is lower than RE, then the NEE is positive, indicating net $CO_2$ sources to the atmosphere. In contrast, negative NEE indicates net $CO_2$ uptake from the atmosphere. The means, standard deviation (SD), and sample sizes for all variables in both restored and control ecosystems were extracted. If some studies (mainly eddy fluxes data) did not include SD values, SD was calculated as 1/10 of the mean[88]. Data in graphical figures and plots were extracted using Web Plot Digitizer (version 4.2).

In this study, we used the definition of "ecological restoration" proposed by the Society for Ecological Restoration. Wetlands have been disturbed by human activities, including the draining of natural wetlands for croplands, grasslands and forests, the conversation of wetlands to aquaculture ponds, peatland extraction, floodplain drainage, mangroves deforestation, etc[3,76]. Wetland restoration is defined as the process of rebuilding the pre-disturbance ecosystem, including the biogeochemical and hydrological processes typical of water-saturated soils, as well as the recovery of vegetation to the former natural ecosystem[29]. According to the collected data in this study, the main types of wetland restoration included the following categories: (1) conversion of drained grasslands to wetlands, (2) conversion of drained croplands to wetlands, (3) conversion of drained forests to wetlands, (4) returning the aquaculture ponds to wetlands, (5) floodplain restoration by rewetting, (6) bog restoration by rewetting and moss layer transfer technique, and (7) mangroves restoration by planting. Based on the collected forest data that meet the selection criteria, forest restoration in this study included the conversion of grasslands to forests and the conversion of croplands to forests[43]. In addition, forest restoration from the disturbances (i.e., clear-cutting, fire and windstorm) was included in the chronosequence sub-dataset (Fig. 5). Grassland degradation was mainly due to overgrazing, land abandonment, or land conversion to croplands[89]. Thus, restoration measures included recovering degraded grassland via grazing exclusion, reducing grazing intensity, artificial assisted restoration, and conversion of croplands to grasslands. The types of grasslands were classified as prairies, temperate steppe & meadow (TGM), alpine steppe & meadow (AGM), and desert steppe (DS)[90].

## Data analysis

We used meta-analysis to examine the response of GHG and environmental factors to ecological restoration. Wetlands, forest and grassland ecosystems can both release $CH_4$ or $CO_2$ into the atmosphere (positive values) and uptake $CH_4$ or $CO_2$ into the atmosphere (negative values). Thus, the natural logarithm-transformed response ratio (RR) is not suitable for meta-analysis in our study. Here, Hedges' d was used to evaluate the weighted response ratios (RRd) as it ranges from $-\infty$ to $+\infty$[43,44]. The Hedges' d was calculated as follows (Eq. 1, Eq. 2):

$$\text{RRd} = \left(1 - \frac{3}{4(N_t + N_c - 2) - 1}\right) \times \frac{X_t - X_c}{S} \tag{1}$$

$$S = \sqrt{\frac{(N_t - 1)S_t^2 + (N_c - 1)S_c^2}{N_t + N_c - 2}} \tag{2}$$

The variance (vd) was estimated with the following equation (Eq. 3):

$$vd = \frac{N_t + N_c}{N_t N_c} + \frac{d^2}{2(N_t + N_c)} \tag{3}$$

where $X_t$ and $X_c$ are the means of the concerned variable in the restored and paired control groups, respectively. S is the pooled standard deviation. $N_t$ and $N_c$ refer to the sample size of variables of the restored and paired control groups, and $S_t$ and $S_c$ are the corresponding standard deviation.

MetaWin 3 software was used to calculate the overall effect size with a categorical random effects model[43]. Confidence intervals (95%; CIs) were calculated by bootstrapping (9999 iterations). The responses of variables to ecological restoration can be considered significant if the 95% confidence interval of the weighted effect size does not overlap zero ($\alpha = 0.05$). If the 95% CI overlapped with zero, it assumed that there was no significant difference between the restored and paired control ecosystems. The effects of ecological restoration on the soil properties also were calculated using the above equations.

To estimate the greenhouse effect of GHG emissions in ecosystems, GWP (t $CO_2$-eq ha$^{-1}$ year$^{-1}$) from $CH_4$, $N_2O$ and $CO_2$ fluxes ($CH_4$ and $CO_2$: g C m$^{-2}$ year$^{-1}$; $N_2O$: g N m$^{-2}$ year$^{-1}$) was calculated as follows (Eq. 4)[1,4,29]:

$$\text{GWP} = \left(27.2 \times \frac{16}{12} \times CH_4 + 273 \times \frac{44}{28} \times N_2O + \frac{44}{12} \times NEE\right) \times 100 \tag{4}$$

where the fractions 16/12, 44/28 and 44/12 were used to transform the mass of C for $CH_4$ and $CO_2$ and nitrogen for the $N_2O$ to $CO_2$ equivalent, respectively[44]. The 27.2 and 273 are the GWP values for $CH_4$ and $N_2O$, respectively, to $CO_2$ over a 100-year time horizon[1].

One-way analysis of variance (ANOVA) was used to test the differences in GHG fluxes ($CH_4$, $N_2O$, GPP, NEE and ER) and soil variables between the restored and paired control ecosystems using IBM SPSS Statistical Tool (Version 23.0; SPSS Inc.). To explore the relationships between GHG fluxes and environmental factors, the mixed meta-regression was adopted after the calculation of the random effect model. In all statistical tests, the level of significance was set at $P < 0.05$.

## Reporting summary

Further information on research design is available in the Nature Portfolio Reporting Summary linked to this article.

## Data availability

All data needed to evaluate the conclusions in the paper are present in the paper and/or the Supplementary Information files. Source data are provided with this paper.

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

## Acknowledgements

This research was supported by the National Natural Science Foundation of China (Nos. 32130069 to X.L.C., 31922060 to K.R.Z., 42007044 to T.H.H., U23A2017 to H.J.X., 32030069 to Q.F.Z. and U1906220 to W.X.D., equally) and Youth Innovation Promotion Association CAS (No. Y2022091 to K.R.Z.).

## Author contributions

K.Z. designed the research, conducted analysis, and wrote the manuscript. T.H., W.D., X.C., Y.Z., Y.C., H.X., X.W., J.Z., K.Z., and Q.Z. collected samples, conducted analysis, and wrote the manuscript. All authors assisted in the discussion of the results and preparation of the manuscript.

## Competing interests

The authors declare no competing interests.
