## [Peer Review File · Nature Communications]

Meta-analysis shows the impacts of ecological restoration on greenhouse gas emissionsREVIEWER COMMENTS

Reviewer #1 (Remarks to the Author):

This study conducted a systematic investigation of the impact of ecosystem restoration on greenhouse gas emissions at a global scale. The magnitude and direction of changes in GHG emissions following ecological restoration of forest, grassland, wetland, and dryland were quantified and potential drivers identified, highlighting the significant potential of ecological restoration to reduce GHG emissions and providing valuable insights and data for improving the IPCC Guidelines for Greenhouse Gas Inventories. Overall, the study is well-written and reports findings that could deepen the understanding of the effect of ecological restoration on GHG emissions. Still, there are a number of issues that must be addressed and the specific comments are listed below:

Abstract

Page 3 Lines 47-50 Ecological restoration on a global scale has a great potential to reduce GHG emissions, so what measures should be prioritized or become more critical for future ecological restoration based on the results of this study? May refer to the last part of the discussion "Implications for the IPCC reports, Guidelines for National Greenhouse Gas Inventories, and future research".

Introduction

Page 7 Lines 107-108 "Some natural wetlands are more efficient carbon sinks than upland terrestrial ecosystems" Why are the above two wetlands being compared? Determine if their definitions are accurate?

Page 7 Lines 107-120 Although the research gap is clearly identified in Lines 121-127, it is not emphasized in the review on page 7, and it is recommended that the current research gap is also emphasized in the review on the impacts of wetland ecosystem restoration on greenhouse gases.

Page 8 Line 132 "(2) explore the patterns of GHG emissions with restoration age" as the

main component of the results, while in the results of “Fig. 5 Changes in the CO₂ fluxes in restored ecosystems with restoration age”, only the results for CO₂ are presented primarily. Could Supplementary Figure 1 Temporal patterns of CH₄ and N₂O flux effect sizes after restoration, be added to Figure 5 of the manuscript results, as a more visual presentation of the results?

Results

Page 11 Line 189 and Table 1: "the C budget" should be defined additionally or a note added to Table 1.

Page 10 Line 178 "Figure. 2c" uses the full spelling and Page 22 Line 442 "Figs. 5" has a clerical error. The formatting should be consistent and the related clerical errors corrected.

Figures 2 and 7: The results presented in Figure 2c only include Grassland to Wetland, Peatland Restoration and Riparian Restoration in Wetland Restoration, what about the rest of the wetland restoration methods? And Figure 7 presents only the NEE of wetland but not the NEE of terrestrial ecosystem?

The results presented in Figure 7 classify forest, grassland, desert, and wetland in this study as wetlands and terrestrial ecosystems, but the results contain less analysis of the comparison of the results of these two types of ecosystems, so it is recommended that they be supplemented. This would also make the focus on the two types of ecosystems and CH₄ emissions in the first part of the discussion logically connect with the results.

Discussion

Page 18 Lines 351-386 The title of this section “Forest, grassland, and wetland restoration enhances C sink and reduces the GWP” includes the role of the three ecosystem restorations, but there is more discussion of forest and wetland restoration, and grassland restoration is largely absent; it is recommended that discussion of the mechanisms associated with grassland restoration that produce the results be added.

Page 20 Lines 387-401 This section states more about the results of this study and discusses

less about comparisons with other studies, suggesting that further references should be added to discuss and explain the reasons for the results.

Reviewer #2 (Remarks to the Author):

The authors conducted a global meta-analysis including 222 peer-reviewed studies on ecosystem restoration, focusing on different habitat types. One of the main findings, amongst others, is that forest, grassland and wetland restoration decreases the global warming potentials (GWP) by respectively 164.5%, 12.6% and 74.5%, by investigation of CH₄, CO₂ and N₂O fluxes. The authors conclude that these numbers can be used for improvement of the IPCC guidelines. Overall, this manuscript is (grammatically) quite well written and it also seems to hold some potentially important numbers and messages.

However, I cannot recommend publication of this work in its current form, which relates to general -but strong- concerns about validity, clarity and context.

Overall, it is of pivotal importance to be much clearer on what “(ecological) restoration” exactly implies, starting already from the abstract/intro onwards. Restoration is a very broad and generic term and this term is rather meaningless without the proper framing and context: it can be interpreted in many different ways, and it is –in practice- done in many different ways. For example, for some ecosystems (e.g. deserts), the meaning of “restoration” is definitely vague and unclear: it is unclear what is meant by “desert restoration”, i.e. desertification is a sign of land degradation and/or climate change (e.g. clearcutting, soil overuse and depletion, global warming,...), and not the other way around. In fact, the concept “desert restoration” is nowhere introduced or discussed, which is problematic as the reader has no idea what is exactly implied here. As a result, the conclusion that “desert restoration” increases CH₄ uptake, “suggesting the great potential of terrestrial ecosystem restoration in enhancing sink function for CH₄”, seems strange.

The same goes for the term “degradation”: what is degradation and what is its context in relation to all of the studied habitat types? Just another example here: there is no such thing as general “grassland degradation”: grasslands can be (negatively) affected by many

different and often contrasting processes, and these processes of interest all need to be clearly defined –one by one- and framed in the paper/intro, with comparison to a reference situation. For example, “grassland degradation” can, amongst others, consist of eutrophication/fertilization, fragmentation, removal of (micro)topography, drainage, encroachment by trees and shrubs, acidification, salinification, pesticide use, and many more, and all of these “degradation processes” are known to have very different and often contrasting outcomes with respect to GHG emissions. The manuscript holds no nuance on that level whatsoever, for none of the habitat types.

Therefore, given the lack of a proper context, delineation, and definition of the words “degradation” and “restoration” (and related processes, which should have been clearly defined) with respect to all of the investigated habitat types, as well as the lack of a proper delineation of the investigated habitat types themselves, including a lack of “references” for restoration (What grassland types (Savannas? Steppes? Calcareous grasslands? Nardus grasslands? Fen meadows?), what wetland types (mangroves? Salt marshes? Bogs? Fens? Floodplains?)), I feel that I cannot properly evaluate nor validate the content of this manuscript. No additional info on that level can be found in the method section either: the method section seems too brief.

In other words; I believe that the manuscript is too generic and too broad, and in my opinion the authors try to cover too many ecosystem types, which results in a loss of detail, nuance and resolution that is in fact required to properly understand the results presented in this paper (this lack of detail and nuance is already very apparent from the intro onwards, e.g. see remark on “desert restoration”). In my opinion, such “try to explain the entire world in one manuscript” approach additionally induces the risk of severe oversimplification of (processes within) the natural environment, which I believe is the case in this manuscript.

Although I acknowledge the potential of (parts of) this work, I believe that it would make more sense to cut this manuscript into smaller separate manuscripts, with more attention to detail, nuance and discussion.

**Responses to reviewers' comments and suggestions**

**REVIEWER COMMENTS**

**Reviewer #1 (Remarks to the Author):**

This study conducted a systematic investigation of the impact of ecosystem restoration
on greenhouse gas emissions at a global scale. The magnitude and direction of changes
in GHG emissions following the ecological restoration of forest, grassland, wetland,
and dryland were quantified and potential drivers identified, highlighting the significant
potential of ecological restoration to reduce GHG emissions and providing valuable
insights and data for improving the IPCC Guidelines for Greenhouse Gas Inventories.
Overall, the study is well-written and reports findings that could deepen the
understanding of the effect of ecological restoration on GHG emissions. Still, there are
a number of issues that must be addressed and the specific comments are listed below:

**Response:** Thank you very much for your comments and suggestions. All the issues
have been addressed in the revised manuscript. According to your suggestions, we have
added the NEE (net ecosystem CO₂ exchange) data and deepened the discussion (Lines
194-202, Lines 391-411).

Abstract

Page 3 Lines 47-50 Ecological restoration on a global scale has a great potential to
reduce GHG emissions, so what measures should be prioritized or become more critical
for future ecological restoration based on the results of this study? May refer to the last
part of the discussion “Implications for the IPCC reports, Guidelines for National
Greenhouse Gas Inventories, and future research”.

**Response:** Thank you for your constructive suggestion. We have added discussions
about the ecological measures in the section “Implications for the IPCC reports,
Guidelines for National Greenhouse Gas Inventories, and future research” in the revised

manuscript.

We have added a paragraph in the Discussion section as follows:

“Our results confirmed that afforestation and reforestation, as well as rewetting
the drained wetlands, should become critical for future ecological restoration to
mitigate GHG (Table S1 and S2, Fig. 5). In addition, previous work reported that the
aquaculture systems are an important source of CH₄ (6.0 Tg CH₄) and N₂O (36.7 Gg
N₂O)⁸⁶, and our results verified that returning aquaculture to wetlands was an effective
measure to mitigate greenhouse gas emissions and enhance CO₂ sinks (Table S1). Bog
restoration by rewetting can be an active restoration strategy to recover vegetation and
convert the extracted bog from GHG sources to GHG sinks²⁶ (Table S2). For the
grassland, restoring the degraded grassland either by grazing exclusion, reducing
grazing intensity, or converting croplands to grassland is an effective strategy for
mitigating GHG (Table S2 and Table S3). These findings offer valuable insights for
policymakers to select effective ecological restoration measures.” (Lines 491-502).

**References**

26. Järveoja, J. et al. Impact of water table level on annual carbon and greenhouse gas
balances of a restored peat extraction area. *Biogeosciences* **13**, 2637–2651

(2016).

86. Yuan, J. et al. Rapid growth in greenhouse gas emissions from the adoption of
industrial-scale aquaculture. *Nat. Clim. Change* **9**, 318–322 (2019).

**Introduction**

Page 7 Lines 107-108 “Some natural wetlands are more efficient carbon sinks than
upland terrestrial ecosystems” Why are the above two wetlands being compared?

Determine if their definitions are accurate?

**Response:** We have revised the sentence considerably in the revised manuscript (Lines
109-112).

We revised the sentences as follows:

“Wetlands are considered to be one of the most efficient ecosystems for
sequestering CO₂ from the atmosphere²⁵, mainly because inundation creates anaerobic

conditions that prevent the decomposition of dead plant material and restore
sequestered C in soil^{26,27}.” (Lines 109-112).

**References**

25. Mitsch WJ, Gosselink JG. 2015 *Wetlands*, 5th edn. Hoboken, NJ: Wiley (2015).

26. Järveoja, J. et al. Impact of water table level on annual carbon and greenhouse gas
balances of a restored peat extraction area. *Biogeosciences* **13**, 2637-2651 (2016).

27. Purre, A.-H., Pajula, R. & Ilomets, M. Carbon dioxide sink function in restored
milled peatlands – The significance of weather and vegetation. *Geoderma* **346**, 30-42
(2019).

Page 7 Lines 107-120 Although the research gap is clearly identified in Lines 121-127,
it is not emphasized in the review on page 7, and it is recommended that the current
research gap is also emphasized in the review on the impacts of wetland ecosystem
restoration on greenhouse gases.

**Response:** Thank you for your constructive suggestion. We have added the impacts of
wetland ecosystem restoration on greenhouse gases in the revised manuscript (Lines
116-127) based on your comment.

We revised the sentences as follows:

“However, the impacts of restoration on wetland GHG and the driving factors
remain controversial³¹⁻³³. Previous work reported that wetland restoration by rewetting
created an anaerobic environment and increased CH₄ emissions by 18 times compared
with the drained wetland, but decreased N₂O and CO₂ emissions, consequently leading
to a net GHG source to the atmosphere^{32,34}. Schaller et al. found that the restored
peatland was still an annual source of CO₂, CH₄ and N₂O after 18 years of rewetting,
resulting in mean net C emissions to the atmosphere³⁵. In contrast, Schrier-Uijl et al.
observed that the enhancement in CO₂ uptake by rewetting former agricultural peatland
outweighed the increase in CH₄ emissions, as well as reduced N₂O emissions, and
therefore turned areas of GHG release into areas of GHG sink 10 years after rewetting³⁶.
The inconsistent results are probably attributed to the restoration age, climate, water
table depth, and soil properties^{32,37}.” (Lines 116-127).

**References**

- 31. IPBES. *The IPBES Assessment Report on Land Degradation and Restoration*
(IPBES Secretariat, 2018) (2018).
- 32. Vanselow-Algan, M. et al. High methane emissions dominated annual greenhouse
gas balances 30 years after bog rewetting. *Biogeosciences* **12**, 4361-4371 (2015).
- 33. Renou-Wilson, F. et al. Rewetting degraded peatlands for climate and biodiversity
benefits: Results from two raised bogs. *Ecol. Eng.* **127**, 547-560 (2019).
- 34. Strack, M. et al. Effect of plant functional type on methane dynamics in a restored
minerotrophic peatland. *Plant Soil* **410**, 231-246 (2016).
- 35 Schaller, C., Hofer, B. & Klemm, O. Greenhouse gas exchange of a NW German
peatland, 18 years after rewetting. *J. Geophys. Res-Bioge.* **127**, 1-21 (2022).
- 36. Schrier-Uijl, A.P. et al. Agricultural peatlands: towards a greenhouse gas sink – a
synthesis of a Dutch landscape study. *Biogeosciences* **11**, 4559-4576 (2014).
- 37. Nugent, K.A., Strachan, I.B., Strack, M., Roulet, N.T. & Rochefort, L. Multi-year
net ecosystem carbon balance of a restored peatland reveals a return to carbon sink.
*Global Change Biol.* **24**, 5751-5768 (2018).

Page 8 Line 132 "(2) explore the patterns of GHG emissions with restoration age" as
the main component of the results, while in the results of "Fig. 5 Changes in the CO₂
fluxes in restored ecosystems with restoration age", only the results for CO₂ are
presented primarily. Could Supplementary Figure 1 Temporal patterns of CH₄ and
N₂O flux effect sizes after restoration, be added to Figure 5 of the manuscript results,
as a more visual presentation of the results?

**Response:** Thank you for your constructive suggestion. We have merged
Supplementary Figure 1 and Fig. 5 according to your comment (Please see the Fig. 5
in the revised manuscript).

**Results**

Page 11 Line 189 and Table 1: "the C budget" should be defined additionally or a note
added to Table 1.

**Response:** We have added it in the revised manuscript.

“C budget, the sum of NEE-C and CH₄-C” (Line 962).

Page 10 Line 178 "Figure. 2c" uses the full spelling and Page 22 Line 442 "Figs. 5" has
a clerical error. The formatting should be consistent and the related clerical errors
corrected.

**Response:** According to your suggestion, we have checked and unified the citation
formats for figures and tables.

Figures 2 and 7: The results presented in Figure 2c only include Grassland to Wetland,
Peatland Restoration and Riparian Restoration in Wetland Restoration, what about the
rest of the wetland restoration methods? And Figure 7 presents only the NEE of wetland
but not the NEE of terrestrial ecosystem?

**Response:** We have added the NEE data of the wetland restoration (converting forest
to wetland, converting cropland to wetland, converting aquaculture to wetland, and
mangrove restoration) and grassland restoration in Figure 2c. In addition, we have
added the NEE data of the grassland in Figure 7. Due to the small samples size for the
paired restored-control measurements for the NEE in forests, the effects of forest
restoration on CO₂ fluxes were not tested by the *RRd* and t-test. The effects of forest
restoration on the NEE were explored by the chronosequence sub-dataset with
restoration age (i.e., years since restoration) (Fig. 5).

**Figure 2c**

**Figure 7**

The results presented in Figure 7 classify forest, grassland, desert, and wetland in this
 study as wetlands and terrestrial ecosystems, but the results contain less analysis of the
 comparison of the results of these two types of ecosystems, so it is recommended that
 they be supplemented. This would also make the focus on the two types of ecosystems

and CH₄ emissions in the first part of the discussion logically connect with the results.
**Response:** In the previous figure 7, we classified forest, grassland, and desert as
terrestrial ecosystems. As suggested by the Reviewer #2, we deleted the desert
ecosystem. Therefore, we reanalyzed each ecosystem separately and remade figure 7
without classifying the wetlands and terrestrial ecosystems in the revised manuscript.
We found that soil bulk density significantly influenced the CH₄ uptake in the grassland
and forest ecosystems while the water table depth significantly affected wetland CH₄
emission. We have revised the discussion of CH₄ emissions to make it logically
coherent in the revised manuscript.

We revised the sentences as follows:

Forest and grassland restoration increased CH₄ uptake while wetland restoration
enhanced CH₄ emissions” (Lines 262-263)

“Unlike the CH₄ emissions in forest and grassland ecosystems which were influenced
by soil properties (Fig.7a, b), the *RRd* of wetland CH₄ emissions was positively
correlated with water table depth, suggesting that the wetland restoration facilitated the
production of CH₄ by creating an anaerobic environment through elevated water table
levels (Figs. 6a, 7c)”. (Lines 307-311)

Discussion

Page 18 Lines 351-386 The title of this section “Forest, grassland, and wetland
restoration enhances C sink and reduces the GWP” includes the role of the three
ecosystem restorations, but there is more discussion of forest and wetland restoration,
and grassland restoration is largely absent; it is recommended that discussion of the
mechanisms associated with grassland restoration that produce the results be added.

**Response:** Thank you for your constructive suggestion. We have added a paragraph to
discuss the grassland restoration in the revised manuscript (Lines 391-411).

We have added a paragraph as follows:

“Grassland restoration markedly reduced the NEE (Fig. 2c), suggesting that
grassland restoration effectively increased C sink capacity. The *RRd* of NEE in

grassland was positively related to the *RRd* of soil moisture (Fig. 7h), indicating that
higher soil moisture increased grassland CO₂ sinks. Grassland restoration by grazing
exclusion or reducing grazing density is conducive to the recovery of grassland and the
increase of vegetation coverage (Fig. 6c), thereby reducing evaporative water loss from
the soil⁷¹. The increase in soil moisture in the restored grassland decreased NEE by
increasing GPP relatively more than ER⁷². The results are consistent with the general
observations that higher soil water content increased vegetation leaf area index and GPP,
and thereby resulted in a great C sink capacity in the grassland ecosystems⁷³. In addition,
the significant relationship between grassland NEE and aboveground biomass
suggested that grassland restoration by grazing exclusion favored the regrowth of
grasses and increased above- and belowground biomass (Fig. 6c), and thus increased
gross ecosystem photosynthesis and eventually resulted in a significant decrease in
NEE⁷⁴. Grassland restoration by converting cropland to grassland remarkably
decreased NEE (Supplementary Fig. S1), indicating an increased C sink. The large
residual of dead roots in the soil with cropland harvest would decompose and release
large CO₂ emissions, while the grassland had a large live root and thereby decreased
ecosystem respiration and increased net CO₂ uptake in the grassland systems⁷⁵.
Therefore, the enhanced CO₂ uptake in the restored grassland was mainly due to the
increase in soil moisture and vegetation biomass (Figs.6c,7h).” (Lines 391-411)

**References**

- 71. Zhu, L., Johnson, D.A., Wang, W., Ma, L. & Rong, Y. Grazing effects on carbon
fluxes in a Northern China grassland. *J. Arid Environ.* **114**, 41-48 (2015).
- 72. Rong, Y., Johnson, D.A., Wang, Z. & Zhu, L. Grazing effects on ecosystem CO₂
fluxes regulated by interannual climate fluctuation in a temperate grassland steppe in
northern China. *Agr. Ecosyst. Environ.* **237**, 194-202 (2017)
- 73. Wang, Y. et al. Carbon fluxes and environmental controls across different alpine
grassland types on the Tibetan Plateau. *Agr. Forest Meteorol.* **311**, 108694 (2021).
- 74. Liu, Y. et al. Grazing exclusion enhanced net ecosystem carbon uptake but
decreased plant nutrient content in an alpine steppe. *Catena* **195**, 104799 (2020).

75. Abraha, M., Hamilton, S.K., Chen, J. & Robertson, G.P. Ecosystem carbon
exchange on conversion of Conservation Reserve Program grasslands to annual and
perennial cropping systems. *Agr. Forest Meteorol.* **253-254**, 151-160 (2018).

Page 20 Lines 387-401 This section states more about the results of this study and
discusses less about comparisons with other studies, suggesting that further references
should be added to discuss and explain the reasons for the results.

**Response:** Based on the suggestion above, we have added some references to deepen
our discussion in the revised manuscript.

We revised the sentences as follows:

“Our results were in accordance with the previous findings that wetland restoration
remarkably decreased GWP to a neutral range (3.3 to 6.7 t CO₂-eq ha⁻¹ yr⁻¹) at the
restored wetlands in comparison with the drained wetlands, mainly due to the increased
CO₂ uptake by precluding oxidation of the residual in the restored wetlands^{29,33,78}.
Afforestation increased CH₄ and CO₂ uptake (Table 1), thus leading to a significant
reduction in GWP with a mean of -10.3 t CO₂-eq ha⁻¹ yr⁻¹. Afforestation provided an
effective strategy for GWP mitigation largely due to the increase in woody biomass C⁸².
Grassland restoration resulted in a decrease in GWP across all the grassland types and
restoration measures (Table 1, Table S2 and S3), mainly by increasing the net CO₂ sink.
Similarly, Rong et al. observed that grassland restoration brought about a greater
increase in GPP than ER in a heavily grazed grassland, therefore decreasing NEE in the
grassland⁷². Taking together, our study suggested that, forest, grassland and wetland
restoration could serve as an effective strategy for mitigating GHG emissions and
reducing GWP.” (Lines 440-453)

**References**

29. Wilson, D. et al. Multiyear greenhouse gas balances at a rewetted temperate
peatland. *Global Change Biol.* **22**, 4080-4095 (2016).

33. Renou-Wilson, F. et al. Rewetting degraded peatlands for climate and biodiversity
benefits: Results from two raised bogs. *Ecol. Eng.* **127**, 547-560 (2019).

72. Rong, Y., Johnson, D.A., Wang, Z. & Zhu, L. Grazing effects on ecosystem CO₂
fluxes regulated by interannual climate fluctuation in a temperate grassland steppe in
northern China. *Agr. Ecosyst. Environ.* **237**, 194-202 (2017).

78. Lee, S.-C. et al. Annual greenhouse gas budget for a bog ecosystem undergoing
restoration by rewetting. *Biogeosciences* **14**, 2799-2814 (2017).

82. Kim, D.-G. & Kirschbaum, M.U.F. The effect of land-use change on the net
exchange rates of greenhouse gases: A compilation of estimates. *Agr. Ecosyst. Environ.*
**208**, 114-126 (2015).

**Reviewer #2 (Remarks to the Author):**

The authors conducted a global meta-analysis including 222 peer-reviewed studies on
ecosystem restoration, focusing on different habitat types. One of the main findings,
amongst others, is that forest, grassland and wetland restoration decreases the global
warming potentials (GWP) by respectively 164.5%, 12.6% and 74.5%, by investigation
of CH₄, CO₂ and N₂O fluxes. The authors conclude that these numbers can be used for
improvement of the IPCC guidelines. Overall, this manuscript is (grammatically) quite
well written and it also seems to hold some potentially important numbers and messages.
However, I cannot recommend publication of this work in its current form, which
relates to general -but strong- concerns about validity, clarity and context.

**Response:** Thank you very much for your comments and suggestions. We tried our best
to address all comments and suggestions in the revised manuscript, especially in terms
of providing a clearer definition of ecological restoration (Lines 67-69) and information
about ecological restoration measures (Lines 560-582). Meanwhile, we have added
some new data and new statistical analyses for exploring the effects of climate factors
(Supplementary Fig. S7), ecological restoration measures (Supplementary Fig. S1), and
ecosystem types (Figs. 2) on GHG emissions in the revised manuscript. We have
deepened our discussion in the revised manuscript.

We revised the discussions as follows:

[revised manuscript text omitted]

Overall, it is of pivotal importance to be much clearer on what “(ecological) restoration”
exactly implies, starting already from the abstract/intro onwards. Restoration is a very
broad and generic term and this term is rather meaningless without the proper framing
and context: it can be interpreted in many different ways, and it is –in practice- done in
many different ways. For example, for some ecosystems (e.g. deserts), the meaning of
“restoration” is definitely vague and unclear: it is unclear what is meant by “desert
restoration”, i.e. desertification is a sign of land degradation and/or climate change (e.g.
clearcutting, soil overuse and depletion, global warming,...), and not the other way
around. In fact, the concept “desert restoration” is nowhere introduced or discussed,
which is problematic as the reader has no idea what is exactly implied here. As a result,
the conclusion that “desert restoration” increases CH₄ uptake, “suggesting the great
potential of terrestrial ecosystem restoration in enhancing sink function for CH₄”, seems
strange.

**Response:** Thank you very much for pointing this out. We deleted the desert restoration
in the revised manuscript according to your suggestion. We have added the definition

of ecological restoration in the introduction (Lines 67-69) and supplemented the types
of different ecological restoration in the part of Methods in the revised manuscript
(Lines 560-582; Fig. 2; Tables S1, S2 and S3). We also have added new statistical
analyses for exploring the effects of climate factors (Supplementary Fig. S7), ecological
restoration measures (Supplementary Fig. S1), and ecosystem types (Fig. 2) on GHG
emissions in the revised manuscript. Meanwhile, we have provided information about
restoration categories and restoration measures in the Supplementary Data.

“Ecological restoration is the process of assisting the recovery of an ecosystem that has
been degraded, damaged, or destroyed (Society for Ecological Restoration and Policy
Working Group 2002).” (Lines 67-69)

“In this study, we used the definition of “ecological restoration” proposed by the
Society for Ecological Restoration.” (Lines 560-561)

The same goes for the term “degradation”: what is degradation and what is its context
in relation to all of the studied habitat types? Just another example here: there is no such
thing as general “grassland degradation”: grasslands can be (negatively) affected by
many different and often contrasting processes, and these processes of interest all need
to be clearly defined –one by one- and framed in the paper/intro, with comparison to a
reference situation. For example, “grassland degradation” can, amongst others, consist
of eutrophication/fertilization, fragmentation, removal of (micro)topography, drainage,
encroachment by trees and shrubs, acidification, salinification, pesticide use, and many
more, and all of these “degradation processes” are known to have very different and
often contrasting outcomes with respect to GHG emissions. The manuscript holds no
nuance on that level whatsoever, for none of the habitat types.

Therefore, given the lack of a proper context, delineation, and definition of the words
“degradation” and “restoration” (and related processes, which should have been clearly
defined) with respect to all of the investigated habitat types, as well as the lack of a
proper delineation of the investigated habitat types themselves, including a lack of

“references” for restoration (What grassland types (Savannas? Steppes? Calcareous
grasslands? Nardus grasslands? Fen meadows?), what wetland types (mangroves? Salt
marshes? Bogs? Fens? Floodplains?)), I feel that I cannot properly evaluate nor validate
the content of this manuscript. No additional info on that level can be found in the
method section either: the method section seems too brief.

**Response:** Thank you very much for the helpful comments and suggestions. In this
study, we used the definition of “ecological restoration” proposed by the Society for
Ecological Restoration and Policy Working Group 2002, i.e., “Ecological restoration is
the process of assisting the recovery of an ecosystem that has been degraded, damaged,
or destroyed (Society for Ecological Restoration and Policy Working Group
2002).” (Lines 67-69). We compiled a global dataset on GHG emissions associated
with ecological restoration. In each study of our dataset, the authors mentioned that
their research systems belong to ecological restoration in their publication. All the
literature in our dataset were peer-reviewed. We have added information about the
restored ecosystems and their paired control ecosystems (i.e., control ecosystems
without restoration treatment), as well as the ecological restoration types in the
Supplementary Data (Please see the Supplementary Data), and showed the results in
figures and tables (Fig. 2, Tables S1, S2 and S3) in our revised manuscript. We have
clearly defined ecological restoration in the Introduction and Methods sections in our
revised manuscript.

We revised the sentences as follows:

“Ecological restoration is the process of assisting the recovery of an ecosystem
that has been degraded, damaged, or destroyed (Society for Ecological Restoration and
Policy Working Group 2002).” (Lines 67-69)

“In this study, we used the definition of “ecological restoration” proposed by the
Society for Ecological Restoration. Wetlands have been disturbed by human activities,
including the draining of natural wetlands for croplands, grasslands and forests, the
conversion of wetlands to aquaculture ponds, peatland extraction, floodplain drainage,
mangroves deforestation, etc.^{3,76}. Wetland restoration is defined as the process of
rebuilding the pre-disturbance ecosystem, including the biogeochemical and

hydrological processes typical of water-saturated soils, as well as the recovery of
vegetation coverage to the former natural ecosystem²⁹. According to the collected data
in this study, the main types of wetland restoration included the following categories:
(1) conversion of drained grasslands to wetlands, (2) conversion of drained croplands
to wetlands, (3) conversion of drained forests to wetlands, (4) returning the aquaculture
ponds to wetland, (5) floodplain restoration by rewetting, (6) bog restoration by
rewetting and moss layer transfer technique, and (7) mangroves restoration by planting.
Based on the collected forest data that meet the selection criteria, forest restoration in
this study included the conversion of grasslands to forests and the conversion of
croplands to forests⁴³. In addition, forest restoration from the disturbances (i.e., clear-
cutting, fire and windstorm) was included in the chronosequence sub-dataset (Fig. 5).
Grassland degradation was mainly due to overgrazing, land abandonment, or land
conversion to croplands⁸⁹. Thus, restoration measures included recovering degraded
grassland via grazing exclusion, reducing grazing intensity, artificial assisted
restoration, and conversion of croplands to grassland. The types of grasslands were
classified as prairies, temperate steppe & meadow (TGM), alpine steppe & meadow
(AGM), and desert steppe (DS)⁹⁰.” (Lines 560-582)

**References**

- 3. Leifeld, J. & Menichetti, L. The underappreciated potential of peatlands in global
climate change mitigation strategies. *Nat. Commun.* **9**, 1071 (2018).
- 29. Wilson, D. et al. Multiyear greenhouse gas balances at a rewetted temperate
peatland. *Global Change Biol.* **22**, 4080-4095 (2016).
- 76. Pu Y.N. The variations of CO₂ and CH₄ fluxes and impact factors in East Lake Taihu
for pre- and post-returning aquaculture to lakes. *Doctoral dissertation*. 2022.
- 89. Bardgett, R.D. et al. Combatting global grassland degradation. *Nat. Rev. Earth Env.*
**2**, 720-735 (2021).
- 90. Wang, J. et al. Vegetation type controls root turnover in global grasslands. *Global*
*Ecol. Biogeogr.* **28**, 442-455 (2019).

In other words; I believe that the manuscript is too generic and too broad, and in my

opinion the authors try to cover too many ecosystem types, which results in a loss of
detail, nuance and resolution that is in fact required to properly understand the results
presented in this paper (this lack of detail and nuance is already very apparent from the
intro onwards, e.g. see remark on “desert restoration”). In my opinion, such “try to
explain the entire world in one manuscript” approach additionally induces the risk of
severe oversimplification of (processes within) the natural environment, which I
believe is the case in this manuscript.

Although I acknowledge the potential of (parts of) this work, I believe that it would
make more sense to cut this manuscript into smaller separate manuscripts, with more
attention to detail, nuance and discussion.

**Response:** Thank you for your constructive suggestion. According to your suggestions,
we have deleted the desert ecosystems. Meanwhile, we have added some new data
compiled from 35 peer-reviewed articles (Please see the Supplementary Data). In order
to deepen the understanding of the effects of ecological restoration on GHG emissions,
we have added new statistical analyses in the revised manuscript as follows: (1) the
relationships of the response ratios (*RRd*) of CH₄, N₂O, and NEE with mean annual
temperature, precipitation, and aridity index (Supplementary Fig. S7), (2) the effects of
ecological restoration measures on CH₄, N₂O, and NEE fluxes across the different
wetland, forest and grassland restoration categories (Supplementary Fig. S1), (3) the
effects of ecological restoration measures and types on CO₂ (GPP and ER) fluxes across
the different wetland, forest and grassland restoration categories (Supplementary Fig.
S2), (4) the effects of ecosystem types on GHG emissions (Fig. 2). Furthermore, we
also further deepened our discussion in the revised manuscript (Please see the Lines
391-411, Lines 440-453, Lines 491-502)

Ecosystem restoration is considered a promising GHG mitigation strategy.
However, a systematic investigation on the impacts of ecosystem restoration on GHG
emissions at a global scale has not yet been conducted. Furthermore, there is a lack of
detailed data about the responses of GHG to ecological restoration in the IPCC reports,
the IPCC Guidelines for National Greenhouse Gas Inventories, and the Good Practice
Guidance for Land Use, Land-Use Change and Forestry. One of the main purposes of

our study is to supplement the available data for the IPCC reports, the IPCC Guidelines
 for National Greenhouse Gas Inventories, and the Good Practice Guidance for Land
 Use, Land-Use Change and Forestry through meta-analysis, thus, three types of
 ecosystems (i.e., wetland, forest and grassland) were included in the current version.

**The newly-added figures and tables were listed as follows:**

**Figure 2 Effects of ecological restoration on CH₄, N₂O, and NEE fluxes across the**
 **different wetland, forest and grassland restoration categories.**

Values are means \pm 95% CIs of the weighted response ratios (*RRd*) between the paired
control ecosystems and restored ecosystems. If the 95% CI value does not overlap
with zero, the response is considered significant. The asterisks indicate significant
effects. Numbers next to the y-axis indicate sample sizes (n). Due to the small sample
size for the paired restored-control measurements for the NEE in forests, the effects of
forest restoration on the NEE were not tested by *RRd*. DG to W, drained grassland to
wetland; DF to W, drained forest to wetland; DC to wetland, drained cropland to
wetland; AQ to Wetland, aquaculture to wetland; NEE, net ecosystem CO₂ exchange;
TGM, temperate steppe & meadow; AGM, alpine steppe & meadow; DS, desert
steppe; AG, artificial grassland.

**Supplementary Figure S1 Effects of ecological restoration measures on CH₄,**

**N₂O, and NEE fluxes across the different wetland, forest and grassland**

**restoration categories.**

Values are means \pm 95% CIs of the weighted response ratios (RRd) between the paired

control ecosystems and restored ecosystems. If the 95% CI value does not overlap
with zero, the response is considered significant. The asterisks indicate significant
effects. Numbers next to the y-axis indicate sample sizes (n). MLTT, moss layer
transfer technique; NR, naturally regeneration; RR, replanting and rewetting; CR to F,
cropland to forest; GR to F; grassland to forest; GE, grazing exclusion; RGD, reduced
grazing density; CR to G, cropland to grassland; AAR, artificial assisted restoration.

**Supplementary Figure S2 Effects of ecological restoration measures (a) and types**
 **(b) on CO₂ (GPP and ER) fluxes across the different wetland, forest and**
 **grassland restoration categories.**

Values are means \pm 95% CIs of the weighted response ratios (*RRd*) between the paired
 control ecosystems and restored ecosystems. If the 95% CI value does not overlap
 with zero, the response is considered significant. The asterisks indicate significant
 effects. Numbers next to the y-axis indicate sample sizes (n). GPP, gross primary
 productivity; ER, ecosystem respiration; MLTT, moss layer transfer technique; NR,
 naturally regeneration; RR, replanting and rewetting; RGD, reduced grazing density;
 AAR, artificial assisted restoration; DG to W; drained grassland to wetland; TGM,
 Temperate steppe & meadow; AGM, Alpine steppe & meadow; DS, Desert steppe.

**Supplementary Figure S7 Relationships of the response ratios (RRd) of CH₄,**
 **N₂O, and NEE with air temperature, precipitation, and aridity index. MAT, mean**
 **annual temperature; MAP, mean annual precipitation. The aridity index is the amount**
 **of average annual precipitation divided by the amount of potential evapotranspiration.**
 **The colored area around the regression line represents the 95% confidence interval,**
 **where n is the number of paired observations.**

**Table S1** Changes in comprehensive C budget and GWP when converting the paired control ecosystems (prefixed with ‘P’) to the restored wetland.

ND, no data; NEE, net ecosystem CO₂ exchange; GWP, global warming potentials; C budget, the sum of NEE-C and CH₄-C

Wetland	CH ₄	N ₂ O	NEE	C budget	GWP	Rate of change
Restoration	kg C ha ⁻¹ year ⁻¹	kg N ha ⁻¹ year ⁻¹	g C m ⁻² year ⁻¹	g C m ⁻² year ⁻¹	t CO ₂ -eq ha ⁻¹ year ⁻¹	%
P-Grasslands	61.2±13.6	5.2±1.4	231.9±93.8	238.0±95.2	12.9±4.5	
Wetlands	284.8±52.1	2.6±0.7	-219.5±62.9	-191.1±68.1	3.4±4.5	-73.8
P-Croplands	3.4±1.9	16.9±7.3	461±183.0	461.3±183.2	24.3±9.9	
Wetlands	182±46.1	2.3±0.6	-140.2±237.5	-121.9±242.1	2.4±10.6	-90.0
P-Forests	9.7±5.2	4.3±2.3	75.6±28.6	76.6±29.1	5.0±2.2	
Wetlands	41.6±15.8	2.2±2.1	10.5±58	14.6±59.6	2.8±3.6	-43.0
P-Aquaculture	225.1±135.1	ND	-41.9±0	-19.4±13.5	6.6±4.9	
Wetlands	165.1±38.2	ND	-151.5±15.6	-135.0±19.4	0.4±2.0	-93.5
P-Floodplains	-2.6±0.4	5.7±5.0	-166.7±84.5	-166.9±84.6	-3.7±5.3	
Restoration	42.8±39.7	3.1±1.9	-265.7±91.9	-261.4±95.8	-6.9±5.6	-82.9
P-Bogs	4.8±1.3	2.2±0.7	159.2±32.5	159.7±32.6	7.0±1.5	
Restoration	92.3±22.8	0.5±0.3	-35.8±25.2	-26.6±27.5	2.2±1.9	-68.0
P-Mangroves	198±0	8.4±2.8	147±0	166.8±0.0	16.2±1.2	
Restoration	215.5±6.5	4.3±1.3	164.8±10.4	186.4±11.1	15.7±1.2	-3.1
Total Control	23.4±6.9	6.7±1.6	176.5±42.1	178.8±42.7	10.2±2.5	
Total Restoration	150.8±17.1	2.1±0.3	-68.5±25.6	-53.4±27.3	3.9±1.7	-62.0

**Table S2** Changes in comprehensive C budget and GWP by different ecological restoration measures. ND, no data; NEE, net ecosystem CO₂
exchange; C budget, the sum of NEE-C and CH₄-C; GWP, global warming potentials; MLTT, moss layer transfer technique; NR, naturally
regeneration; RR, replanting and rewetting; RGD, reduced grazing density; GE, grazing exclusion; AAR, artificial assisted restoration

Ecosystem	Restoration type	CH ₄ kg C ha ⁻¹ year ⁻¹	N ₂ O kg N ha ⁻¹ year ⁻¹	NEE g C m ⁻² year ⁻¹	C budget g C m ⁻² year ⁻¹	GWP t CO ₂ -eq ha ⁻¹ year ⁻¹	Rate of change %	
Wetland	Control	18.7±5.7	5.8±1.3	217.4±56.1	219.2±56.7	11.1±2.8		
	Rewetting	197.2±27.8	2.0±0.4	-129.4±38.5	-109.6±41.3	3.3±2.6	-70.4	
	Control	2.3±0.9	1.6±0.7	299.2±31.4	299.4±31.4	11.8±1.5		
	MLTT	21.9±4.8	0.01±0.1	116.6±87.5	118.7±88.0	5.1±3.4	-56.8	
	Control	5.4±1.4	NA	23.8±1.8	24.3±1.9	1.1±0.1		
	NR	65.3±45.2	NA	-126.8±12.5	-120.2±17.0	-2.3±2.1	-313.7	
	Control	198.0±0	2.7±1.2	147±0.0	166.8±0.0	13.7±0.5		
	Replanting	227.6±6.7	1.1±0.7	164.8±10.4	187.6±11.1	14.8±0.9	7.6	
Forest	Control	12.9±6.3	17.1±11.4	64.6±8.8	65.9±9.5	10.2±5.5		
	RR	55.9±8.5	4.0±1.3	-110.4±43.0	-104.9±43.8	-0.3±2.5	-102.9	
	P-Croplands	-1.3±0.3	3.7±0.9	-501.3	-501.4	-16.9±0.4		
	Forests	-2.5±0.2	1.4±0.3	-957.8	-957.1	-34.6±0.1	-105.4	
	P-Grasslands	-0.7±0.5	0.5±0.2	15.7±90.1	15.6±90.1	0.8±3.4		
	Forests	-1.4±0.3	1.4±0.4	-129.5±105.5	-129.7±105.5	-4.2±4.0	-651.8	
	Grassland	Control	-3.8±0.8	0.3±0.1	-587.9±119.1	-588.3±119.2	-21.6±4.4	
		RGD	-5.5±0.8	0.2±0.0	-1460.1±463.0	-1460.6±463.1	-53.7±17.0	-148.8
Control		-2.6±0.4	0.6±0.2	-245.3±94.0	-245.6±94.1	-8.8±3.6		
GE		-3.3±0.5	0.6±0.2	-703.0±188.4	-703.3±188.4	-25.6±7.0	-190.5	
Control		-2.5±0.8	1.2±0.5	96.9±363.6	96.6±363.7	4.0±13.6		
AAR		-3.2±0.8	2.5±0.4	-255.4±402.9	-255.7±403.0	-8.4±15.0	-312.2	
Control		-0.8±0.4	2.3±0.7	10.3±30.1	10.2±30.2	1.3±1.4		
C to R	-0.9±0.2	0.7±0.3	-75.8±20.1	-75.9±20.1	-2.5±0.9	-289.1		

**Table S3** Changes in comprehensive C budget and GWP in different grassland types. ND, no data; NEE, net ecosystem CO₂ exchange; C budget,
 the sum of NEE-C and CH₄-C; GWP, global warming potentials; TGM, Temperate steppe & meadow; AGM, Alpine steppe & meadow

Grassland type	CH ₄ kg C ha ⁻¹ year ⁻¹	N ₂ O kg N ha ⁻¹ year ⁻¹	NEE g C m ⁻² year ⁻¹	C budget g C m ⁻² year ⁻¹	GWP t CO ₂ -eq ha ⁻¹ year ⁻¹	Rate of change %
Control	-2.6±0.4	0.5±0.3	-446.7±111.2	-447±111.3	-16.2±4.2	
TGM	-3.8±0.4	0.6±0.2	-1013.1±241.4	-1013.5±241.4	-37.0±8.9	-128.1
Control	-0.03±0.03	4.8±0.6	NA	NA	2.0±0.2	
Prairie	0.0035±0.1	0.1±0.1	NA	NA	0.1±0.0	-97.2
Control	-7.7±0	NA	61.0±36.3	60.2±36.3	2.0±1.3	
Desert steppe	-11.4±0.4	NA	-144.9±112.8	-146.1±112.9	-5.7±4.2	-392.8
Control	-1.7±0.3	1.0±0.5	NA	NA	0.4±0.2	
Artificial grassland	-1.7±0.2	0.5±0.1	NA	NA	0.2±0.1	-56.7
Control	-1.9±0.9	0.6±0.3	-162.7±187.8	-162.8±187.9	-5.8±7.0	
AGW	-2.5±0.6	1.0±0.3	-726.6±311.3	-726.8±311.4	-26.3±11.6	-354.6

536

REVIEWER COMMENTS

Reviewer #1 (Remarks to the Author):

The author has made sufficient modifications, but there are still some issues that need to be improved:

The abstract states that the implications of the study include "valuable insights and new data for improving the IPCC Guidelines for Greenhouse Gas Inventories". This description is too general, are there specific insights for improving the IPCC Guidelines for Greenhouse Gas Inventories?

While the introduction discusses in detail the current researches in forest, grassland, and wetland ecosystems, these three ecosystems are not highlighted in the summary of research gaps in current studies. So it is suggested to add more information to correspond with the previous content. This section is also written in a slightly repetitive manner by looking at the deficiencies in each of the three ecosystems, and I suggest the authors focus on the research gaps in ecological restoration and greenhouse gas emissions in the three ecosystem types.

Lines 139-140 "(2) explore the patterns of GHG emissions with restoration age, and (3) determine the key factors influencing the response of GHG emissions to ecological restoration." Restoration age is also one of the factors influencing GHG emissions, and the results of the patterns of GHG emissions with restoration age and other influencing factors are both presented in the section "Factors influencing the response of CH₄, N₂O, and NEE to ecological restoration". Is it possible to consider further emphasizing the results of the section on restoration age? And explain why the restoration age is separately included as a part of the results, and whether this part of the results is of more significance.

Although the changes of wetlands are added in the results, they are not comprehensively analyzed and compared. The features of this manuscript are not highlighted.

Reviewer #2 (Remarks to the Author):

Dear editor,

In my first review, I was primarily concerned about the lack of much needed context and detail on the exact meaning of "ecological restoration" with respect to different measures that can be taken in the field, and across different ecosystems.

It seems that the authors have now provided a lot more clarity on this matter, as well as additional suppl. figures that show more detail, as requested. The topic of "desert restoration" was fully removed, which I think was necessary as well.

Overall, these changes imply that my main concerns with the initial manuscript have now been resolved.

Sincerely,

[EDITORIAL NOTE: Reviewer name is redacted as they do not wish to be named]

Point-by-Point Response to the Reviewers' Comments

REVIEWER COMMENTS

Reviewer #1 (Remarks to the Author):

The author has made sufficient modifications, but there are still some issues that need to be improved:

Response:

Thank you very much for your comments and suggestions. These suggestions are very helpful for us to improve the quality of our manuscript. All the issues have been addressed in the revised manuscript. Please see the below response to each point.

The abstract states that the implications of the study include "valuable insights and new data for improving the IPCC Guidelines for Greenhouse Gas Inventories". This description is too general, are there specific insights for improving the IPCC Guidelines for Greenhouse Gas Inventories?

Response:

Thank you for your constructive suggestion. We have added some specific insights in the revised manuscript as follows:

“Our findings underscore the vast potential of ecological restoration in mitigating GHG emissions at a global scale, provide valuable insights and new data for improving the IPCC Guidelines for Greenhouse Gas Inventories, and suggest that afforestation, reforestation, rewetting the drained wetlands, and restoring the degraded grassland through grazing exclusion, reducing grazing intensity, or converting cropland to grassland can effectively mitigate GHG emissions.” (Lines 47-52).

While the introduction discusses in detail the current researches in forest, grassland, and wetland ecosystems, these three ecosystems are not highlighted in the summary of research gaps in current studies. So it is suggested to add more information to correspond with the previous content. This section is also written in a slightly

repetitive manner by looking at the deficiencies in each of the three ecosystems, and I suggest the authors focus on the research gaps in ecological restoration and greenhouse gas emissions in the three ecosystem types.

Response:

Thank you for your constructive suggestion. We have improved the introduction and focused on the research gaps in ecological restoration and greenhouse gas emissions in the three ecosystem types. We revised the sentences as follows:

“Although many studies showed that afforestation could enhance the CO₂ sink function of ecosystems^{10,14}, some studies observed that forest lands continued to act as a CO₂ source even after several years of afforestation¹⁵.” (Lines 90-93)

“Previous work reported that grassland restoration increased C accumulation and enhanced CH₄ uptake²⁴, but some studies found that grassland restoration might stimulate N₂O and CO₂ emissions and shift grassland from a C sink to a C source¹⁹. Furthermore, the effects of grassland types, restoration measures, and restoration age on GHG emissions in the restored grasslands at a global scale are still unclear.” (Lines 104-109)

“Previous work reported that wetland restoration could shift the ecosystems into net GHG sources^{32,34,35} or net sinks³⁶. The inconsistent results are probably attributed to the wetland restoration types, restoration age, climate, water table depth, and soil properties^{32,37}. Since wetland restoration generally decreases CO₂ emissions but increases CH₄ emissions^{32,34,36}, the overall effects of wetland restoration on the global warming potentials (GWP) considering three major GHGs (i.e., CO₂, CH₄, and N₂O) are not well understood.” (Lines 118-125)

“Furthermore, there is currently a lack of comprehensive global assessments for the three major ecosystems (i.e., forests, grasslands, and wetlands) which are crucial for the global GHG budget and the ‘UN Decade on Ecosystem Restoration’^{6,7,8,9}.” (Lines 129-132)

References

6. Girardin, C.A. et al. Nature-based solutions can help cool the planet-if we act now.

- Nature* **593**, 191–194 (2021).
7. Schimel, D.S. et al. Recent patterns and mechanisms of carbon exchange by terrestrial ecosystems. *Nature* **414**, 169–172 (2001).
 8. Akande, O.J., Ma, Z., Huang, C., He, F. & Chang, S.X.J.E.L. Meta-analysis shows forest soil CO₂ effluxes are dependent on the disturbance regime and biome type. *Ecol. Lett.* **26**, 765–777 (2023).
 9. Feng, H. et al. Global estimates of forest soil methane flux identify a temperate and tropical forest methane sink. *Geoderma* **429**, 116239 (2023).
 10. Harris, N.L. et al. Global maps of twenty-first century forest carbon fluxes. *Nat. Clim. Change* **11**, 234–240 (2021).
 14. Cai, T., Price, D.T., Orchansky, A.L. & Thomas, B.R. Carbon, water, and energy exchanges of a hybrid poplar plantation during the first five years following planting. *Ecosystems* **14**, 658–671 (2011).
 15. Peichl, M., Arain, M.A. & Brodeur, J.J. Age effects on carbon fluxes in temperate pine forests. *Agr. Forest Meteorol.* **150**, 1090–1101 (2010).
 19. Hortnagl, L. et al. Greenhouse gas fluxes over managed grasslands in Central Europe. *Global Change Biol.* **24**, 1843–1872 (2018).
 24. Liu, Z. et al. Grassland restoration measures alter soil methane uptake by changing community phylogenetic structure and soil properties. *Ecol. Indic.* **133**, 108368 (2021).
 32. Vanselow-Algan, M. et al. High methane emissions dominated annual greenhouse gas balances 30 years after bog rewetting. *Biogeosciences* **12**, 4361–4371 (2015).
 34. Strack, M. et al. Effect of plant functional type on methane dynamics in a restored minerotrophic peatland. *Plant Soil* **410**, 231–246 (2016).
 - 35 Schaller, C., Hofer, B. & Klemm, O. Greenhouse gas exchange of a NW German peatland, 18 years after rewetting. *J. Geophys. Res-Biogeophys.* **127**, 1–21 (2022).
 36. Schrier-Uijl, A.P. et al. Agricultural peatlands: towards a greenhouse gas sink – a

synthesis of a Dutch landscape study. *Biogeosciences* **11**, 4559–4576 (2014).

37. Nugent, K.A., Strachan, I.B., Strack, M., Roulet, N.T. & Rochefort, L. Multi-year net ecosystem carbon balance of a restored peatland reveals a return to carbon sink. *Global Change Biol.* **24**, 5751–5768 (2018).

Lines 139-140 "(2) explore the patterns of GHG emissions with restoration age, and (3) determine the key factors influencing the response of GHG emissions to ecological restoration." Restoration age is also one of the factors influencing GHG emissions, and the results of the patterns of GHG emissions with restoration age and other influencing factors are both presented in the section "Factors influencing the response of CH₄, N₂O, and NEE to ecological restoration". Is it possible to consider further emphasizing the results of the section on restoration age? And explain why the restoration age is separately included as a part of the results, and whether this part of the results is of more significance.

Response:

Thank you for your constructive suggestion. We have added a section in the results to further emphasize the restoration age with the subtitle “Changes of CH₄ and N₂O emissions and NEE with restoration age”. All the results related to the restoration age in the previous version have been moved to this section. In addition, we have further emphasized the importance of the restoration age in the discussion section.

We revised the sentences as follows:

Results section

“Changes of CH₄ and N₂O emissions and NEE with restoration age

Given the critical impact of restoration age on GHG emissions in restored ecosystems, the patterns of CH₄ and N₂O emissions and NEE with restoration age were first explored.” (Lines 219-222)

Discussion section

“Our study highlights the significance of restoration age in regulating GHG emissions in restored ecosystems, underscoring the importance of considering the time in assessing or modeling the effects of restoration or land-use change on GHG emissions (Fig. 5 and Supplementary Fig. S3). Although restoration measures can be implemented and completed quickly, the re-establishment of plant coverage and microbial communities is a gradual process^{50,77,78}. The process of biomass accumulation changes over time^{15,77}, and soil physical, biogeochemical, and hydrological properties change with restoration time^{39,41,50}. Consequently, restoration age plays a significant role in regulating GHG budgets. (Lines 534-542)

References

15. Peichl, M., Arain, M.A. & Brodeur, J.J. Age effects on carbon fluxes in temperate pine forests. *Agr. Forest Meteorol.* **150**, 1090-1101 (2010).
39. Hiltbrunner, D., Zimmermann, S., Karbin, S., Hagedorn, F. & Niklaus, P.A. Increasing soil methane sink along a 120-year afforestation chronosequence is driven by soil moisture. *Global Change Biol.* **18**, 3664-3671 (2012).
41. Baah-Acheamfour, M., Carlyle, C.N., Lim, S.S., Bork, E.W. & Chang, S.X. Forest and grassland cover types reduce net greenhouse gas emissions from agricultural soils. *Sci. Total Environ.* **571**, 1115-1127 (2016).
50. Bárcena, T.G. et al. Conversion of cropland to forest increases soil CH₄ oxidation and abundance of CH₄ oxidizing bacteria with stand age. *Appl. Soil Ecol.* **79**, 49-58 (2014).
77. Waddington, J.M., Strack, M. & Greenwood, M.J. Toward restoring the net carbon sink function of degraded peatlands: Short-term response in CO₂ exchange to ecosystem-scale restoration. *J. Geophys. Res.* **115** G1 (2010).
78. Lee, S.-C. et al. Annual greenhouse gas budget for a bog ecosystem undergoing restoration by rewetting. *Biogeosciences* **14**, 2799-2814 (2017).

Although the changes of wetlands are added in the results, they are not comprehensively analyzed and compared. The features of this manuscript are not highlighted.

Response:

Thank you for your constructive suggestion. We have comprehensively analyzed and compared these results and highlighted the features in the revised manuscript as follows:

“Since wetland restoration generally decreases CO₂ emissions but increases CH₄ emissions^{32,34,36}, the overall effects of wetland restoration on the global warming potentials (GWP) considering three major GHGs (i.e., CO₂, CH₄, and N₂O) are not well understood.” (Line 122-125)

“Furthermore, there is currently a lack of comprehensive global assessments for the three major ecosystems (i.e., forests, grasslands, and wetlands) which are crucial for the global GHG budget and the ‘UN Decade on Ecosystem Restoration’^{6,7,8,9}.” (Line 129-132)

“Among the types of wetland restoration, the conversion of grasslands to wetlands showed the largest increase in CH₄ emissions, followed by the conversion of croplands to wetlands (Fig. 3a). These results indicated that greater attention should be paid to the increased CH₄ emissions from the restored wetlands in global GHG accounting.” (Lines 317-321)

“Bog restoration by rewetting may be beneficial to the proliferation of aerenchymatous vascular plants, and thus allow CH₄ to bypass the oxidized surface soil, consequently enhancing CH₄ emission into the atmosphere via the plant-mediated transport³².” (Lines 331-334)

“Generally, wetland restoration could reduce GWP by 43-90% (Table S1). Among the wetland restoration measures, rewetting, moss layer transfer, and replanting & rewetting could significantly decrease GWP (Table S2). Returning croplands to wetlands is the most effective way to reduce GWP (Table S1).” (Lines 462-465)

“Previous work found that the restoration of bogs through rewetting could yield exceptionally high CO₂ sink, which could effectively offset CH₄ emissions, thus resulting in a substantial reduction in the GWP²⁹.” (Lines 470-472)

“Wetland restoration by converting drained forests, grasslands, and croplands into wetlands increased CO₂ sinks by decreasing ecosystem respiration, effectively transforming the ecosystems from CO₂ sources to sinks²⁹.” (Lines 525-527)

References

6. Girardin, C.A. et al. Nature-based solutions can help cool the planet-if we act now. *Nature* **593**, 191–194 (2021).
7. Schimel, D.S. et al. Recent patterns and mechanisms of carbon exchange by terrestrial ecosystems. *Nature* **414**, 169–172 (2001).
8. Akande, O.J., Ma, Z., Huang, C., He, F. & Chang, S.X.J.E.L. Meta-analysis shows forest soil CO₂ effluxes are dependent on the disturbance regime and biome type. *Ecol. Lett.* **26**, 765–777 (2023).
9. Feng, H. et al. Global estimates of forest soil methane flux identify a temperate and tropical forest methane sink. *Geoderma* **429**, 116239 (2023).
29. Wilson, D. et al. Multiyear greenhouse gas balances at a rewetted temperate peatland. *Global Change Biol.* **22**, 4080–4095 (2016).
32. Vanselow-Algan, M. et al. High methane emissions dominated annual greenhouse gas balances 30 years after bog rewetting. *Biogeosciences* **12**, 4361–4371 (2015).
34. Strack, M. et al. Effect of plant functional type on methane dynamics in a restored minerotrophic peatland. *Plant Soil* **410**, 231–246 (2016).
36. Schrier-Uijl, A.P. et al. Agricultural peatlands: towards a greenhouse gas sink – a synthesis of a Dutch landscape study. *Biogeosciences* **11**, 4559–4576 (2014).

Reviewer #2 (Remarks to the Author):

Dear editor,

In my first review, I was primarily concerned about the lack of much needed context and detail on the exact meaning of "ecological restoration" with respect to different measures that can be taken in the field, and across different ecosystems.

It seems that the authors have now provided a lot more clarity on this matter, as well as additional suppl. figures that show more detail, as requested. The topic of "desert restoration" was fully removed, which I think was necessary as well.

Overall, these changes imply that my main concerns with the initial manuscript have now been resolved.

Sincerely,

[EDITORIAL NOTE: Reviewer name is redacted as they do not wish to be named]

Response:

We are grateful to you for the constructive suggestions. Your suggestions are of great help in improving the quality of our manuscript. Thank you so much.

REVIEWERS' COMMENTS

Reviewer #1 (Remarks to the Author):

The author has revised the article sufficiently to clarify the introduction and also to add more details about the results of the patterns of GHG emissions with restoration age. More details have been added based on the suggestions.

Overall, the author's revision has addressed all of my comments.